# TransPL: VQ-Code Transition Matrices for Pseudo-Labeling of Time Series Unsupervised Domain Adaptation

**Jaeho Kim** [1]   **Seulki Lee** [1]

## Abstract

Unsupervised domain adaptation (UDA) for time series data remains a critical challenge in deep learning, with traditional pseudo-labeling strategies failing to capture temporal patterns and channel-wise shifts between domains, producing sub-optimal pseudo labels. As such, we introduce TransPL, a novel approach that addresses these limitations by modeling the joint distribution $\mathcal{P}(\mathbf{X}, y)$ of the source domain through code transition matrices, where the codes are derived from vector quantization (VQ) of time series patches. Our method constructs class- and channel-wise code transition matrices from the source domain and employs Bayes' rule for target domain adaptation, generating pseudo-labels based on channel-wise weighted class-conditional likelihoods. TransPL offers three key advantages: explicit modeling of temporal transitions and channel-wise shifts between different domains, versatility towards different UDA scenarios (*e.g.* weakly-supervised UDA), and explainable pseudo-label generation. We validate TransPL's effectiveness through extensive analysis on four time series UDA benchmarks and confirm that it consistently outperforms state-of-the-art pseudo-labeling methods by a strong margin (6.1% accuracy improvement, 4.9% F1 improvement), while providing interpretable insights into the domain adaptation process through its learned code transition matrices.

## 1. Introduction

Time series often exhibit strong variability across different domains (*e.g.* users, devices) due to both dynamic temporal transitions and inherent sensor characteristics. As such, a model trained on one specific domain (*i.e.* source) may not generalize to previously unseen domains (*i.e.* target). For instance, in human activity recognition using wearable devices, the accelerometer's gravity component may remain stable across users but the gyroscope readings can shift dramatically when users wear devices with varying tightness (Parkka et al., 2007). Therefore, **characterizing the temporal transitions within time series and incorporating selective shift in sensors (channels) for time series** domain adaptation potentially enhances both the model adaptation process and the transparency of the black-box methodology, as we understand "what" needs to be adapted.

In this work, we propose TransPL, a novel unsupervised domain adaptation (UDA) pseudo-labeling strategy for time series classification. TransPL models the temporal transitions via vector quantized (VQ) codebooks (Van Den Oord et al., 2017) and applies a selective channel adaptation strategy by quantifying the alignment between the channel code transition matrices of the source and target domains. In the UDA setup, the target model has access to both the labeled source and the unlabeled target domain training sets. TransPL models the joint $\mathcal{P}(\mathbf{X}, y)$ of the labeled source domain through the construction of code transition matrices, where the transition matrices are used for pseudo-labeling the unlabeled target's training set using Bayes' rule. Previous pseudo-labeling strategies (He et al., 2023b) are mostly derived from image domains (Lee et al., 2013; Liu et al., 2023b), where they operate as black boxes that fail to explicitly model the temporal dynamics and selective channel shifts inherent to time series. Instead, they rely on the source domain's classifier or the clustering of the target domain, resulting in sub-par model performance and black-box pseudo-labels. In light of this, TransPL provides the following contributions:[1]

1. We formulate unsupervised domain adaptation (UDA) for time series as learning discrete transitions between vector quantized (VQ) codes that characterize temporal dynamics based on our novel coarse and fine VQ structure. The learned codes are used to construct class-

---

[1] Artificial Intelligence Graduate School, Ulsan National Institute of Science and Technology (UNIST), Ulsan, South Korea. Correspondence to: Seulki Lee <seulki.lee@unist.ac.kr>.

*Proceedings of the 42$^{nd}$ International Conference on Machine Learning*, Vancouver, Canada. PMLR 267, 2025. Copyright 2025 by the author(s).

[1] https://github.com/eai-lab/TransPL

and channel-wise transition matrices.

2. Using Bayes' rule, we calculate the class posteriors in a channel-wise manner, where the posteriors are calculated from the class-conditional likelihoods and a prior. This widens TransPL's applicability to various UDA setups, including weakly supervised scenarios, where the label distribution (*i.e.* prior) is known. Furthermore, we quantify channel alignment scores using the optimal transport (Flamary et al., 2021), allowing us to weight channel-wise confidence of the class posteriors.

3. We demonstrate that TransPL outperforms existing pseudo-labeling strategies, resulting in enhanced UDA performance across time series benchmarks, as well being interpretable, providing transparent explanations for the generated pseudo-labels.

## 2. Backgrounds and Related Works

### 2.1. Pseudo Labeling in UDA

Pseudo labeling constructs synthetic class labels (*i.e.* pseudo labels) for unlabeled target domain samples in unsupervised domain adaptation (UDA), where the labels serve as a supervision signal for training the model on target domain data. Existing pseudo labeling approaches can be classified into several sub-types. Confidence-based methods such as Softmax (Lee et al., 2013) utilize prediction probabilities from source-trained classifiers; ATT (Saito et al., 2017) uses multiple classifiers to obtain pseudo labels through prediction agreement. Prototype-based methods like nearest class prototype (NCP) (Wang & Breckon, 2020) leverage learned source class prototype representations. Another category includes clustering-based techniques that rely on the latent representation of unlabeled target domains (Wang & Breckon, 2020). These methods have been further enhanced through hybrid approaches that combine clustering with confidence-based filtering to identify and remove unreliable pseudo labels, as demonstrated in SHOT (Liang et al., 2020) and T2PL (Liu et al., 2023a). However, these methodologies have mostly been developed for static data like images, and their application to time series (He et al., 2023b) is suboptimal as they fail to capture two critical aspects: the temporal dynamics inherent in sequential data and the multi-channel characteristics of time series. Unlike images, time series exhibit strong temporal dependencies and channel-wise shifts that require explicit modeling for effective domain adaptation. Our TransPL bridges this gap for the first time by explicitly modeling both the temporal transitions and channel-wise shifts via vector quantized (VQ) codebooks and transition matrices.

### 2.2. Unsupervised Domain Adaptation in Time Series

Due to the lack of semantics in time series, it is often known that labeling time series is costly compared to images and text (Kim et al., 2024). As such, the UDA setup is particularly important for time series, where obtaining labeled data from different domains (*e.g.* users) is expensive. CoDATS (Wilson et al., 2020) is an extension of DANN (Ganin et al., 2016), a domain adversarial approach that aligns source and target through adversarial training. Notably, CoDATS introduced the concept of weak supervision (WS) in time series through target label distributions, which is especially relevant in practical scenarios such as activity recognition where users can provide self-reported estimates of time spent on different activities. Here, the WS in CoDATS is provided through an ad-hoc minimization of Kullback-Leibler (KL) divergence between the expectation of the target prediction distribution and the weakly-supervised label distribution. In contrast, TransPL can integrate WS directly into the pseudo label formulation through Bayes' rule, providing a mathematically principled approach to incorporating prior knowledge of target label distributions.

More recent approaches have explored various aspects of time series adaptation. SASA (Cai et al., 2021) constructs and aligns sparse associative structures between domains using attention maps. RAINCOAT (He et al., 2023a) leverages both time and frequency features for domain alignment, arguing that frequency features exhibit stronger domain invariance. CauDiTS (Sun et al., 2024) approaches the problem through causality, disentangling causal and non-causal patterns in time series, where causal patterns maintain domain invariance. However, none of these methods explicitly model temporal state transitions or channel-wise shifts, nor do they provide an explainable outcome of the adaptation process, which are key contributions of TransPL.

After our submission, we discovered SSSS-TSA (Ahad et al., 2025) addresses channel-level variations using self-attention mechanisms to select relevant channels for alignment. While we share similar motivations, SSSS-TSA differs from ours by employing self-attention for channel selection and implementing separable channel-level encoders and classifiers, where there is a clear methodological distinction in how channels are selectively aligned.

### 2.3. Time Series Discretization

Methods like Symbolic approXimation (SAX) (Lin et al., 2003) transform continuous time series into symbolic sequences by discretizing amplitude values into a finite alphabet. However, the transformation typically relies on predefined quantization schemes that may not capture data-specific patterns. VQVAE (Vector Quantized Variational Auto Encoder) (Van Den Oord et al., 2017) is a generative model that learns discrete representations (*i.e.* codes) of continuous data through a codebook-based quantization approach. VQVAE learns the discretization step directly from data, allowing for more adaptive and meaningful representa-

tions of temporal patterns. In time series, VQVAE has been used for generation (Lee et al., 2023), self-supervised learning (Gui et al., 2024), and for foundation models (Talukder et al., 2024). Here, TransPL is the first to leverage VQVAE for UDA in time series and to provide explainable insights through the use of code transition matrices.

## 3. Overview

In this section, we formalize the notations, and provide an overview of TransPL with an illustrative figure in Figure 1.

**Notations.** Formally, we define the labeled source domain as $\mathcal{S} = \{(\mathbf{X}, y) | \mathbf{X} \sim \mathcal{P}_X^{\mathcal{S}}, y \sim \mathcal{P}_Y^{\mathcal{S}}\}$, where $\mathcal{P}_X$ and $\mathcal{P}_Y$ are the instance and label distributions. Let $\mathbf{X} \in \mathbb{R}^{D \times T}$ be a time series instance, where $D$ and $T$ denote the number of channels (sensors) and sequence length, respectively. We employ the superscript $d = 1, ..., D$ to refer to each individual channel $\mathbf{X}^d$. The unlabeled target domain $\mathcal{T} = \{\mathbf{X} | \mathbf{X} \sim \mathcal{P}_X^{\mathcal{T}}\}$, is accessible during model training. Following the covariate shift assumption (Zhang et al., 2013), we assume $\mathcal{P}(y|\mathbf{X})^{\mathcal{S}} = \mathcal{P}(y|\mathbf{X})^{\mathcal{T}}$ while $\mathcal{P}_X^{\mathcal{S}} \neq \mathcal{P}_X^{\mathcal{T}}$. Here, the label spaces $\mathcal{P}_Y^{\mathcal{S}}$ and $\mathcal{P}_Y^{\mathcal{T}}$ are identical, with $y \in \{1, ..., K\}$, and $K$ denoting the total number of classes.

**Overview.** We first segment the time series into non-overlapping patches (*i.e.* tokens) of length $m$, where we obtain $N = \lfloor \frac{T}{m} \rfloor$ number of patches for each channel. As such, we can reshape the time series instance $\mathbf{X} \in \mathbb{R}^{D \times T}$ into $\mathbf{X}^{\text{re}} \in \mathbb{R}^{D \times N \times m}$, and encode each patches into latent vectors using a patch encoder $\mathcal{E}_\theta$ (*i.e.* Transformer). Patchifying time series enables each temporal segment to be mapped to discrete vector quantized (VQ) codes, which are later used for the construction of coarse code transition matrices, allowing us to represent temporal transitions and channel-wise alignment explicitly. Section 4 details the mapping of latent patches to discrete VQ codes and their utilization in constructing three different types of transition matrices (TM): the class-wise TM from the source domain, and channel-wise TM from both source and target domains. Section 5 describes how the class-wise TM is used to construct class-conditional likelihoods for the unlabeled target sequence, while the channel-wise TMs are used for calculating channel alignment scores using optimal transport (Flamary et al., 2021). These likelihoods and channel-wise alignment guide the generation of pseudo labels for the unlabeled target domain, where the model is fine-tuned using the labeled source and pseudo-labeled target data for domain adaptation.

## 4. Source Training

The objective of source training is to model the joint $\mathcal{P}(\mathbf{X}, y)^{\mathcal{S}}$ from the source by constructing meaningful vector quantized (VQ) code sequences from the labeled source data. Directly modeling the joint distribution from

raw time series data is impractical due to its high dimensionality and continuous nature, which makes density estimation in the raw input space challenging. Therefore, we use a patch encoder $\mathcal{E}_\theta$ (*i.e.* Transformer) as the backbone to encode patches $\mathbf{X}^{\text{re}} \in \mathbb{R}^{D \times N \times m}$ into a latent representation $\mathbf{Z} \in \mathbb{R}^{D \times N \times d_{\text{dim}}}$, where $d_{\text{dim}}$ is the size of the latent. To simplify the notation in the rest of this paper, we will focus on a single channel $d$ where the latent patch sequence is given as $\mathbf{Z}^{[d,:,:]} = [\mathbf{z}_1, ..., \mathbf{z}_N]$. We then employ VQ to map the patches into discrete codes, reducing the complexity while preserving temporal dynamics. This discretization enables modeling the joint distribution through tractable transition matrices from VQ codes. Here, the question is how to guide the VQ codes to capture generalized temporal patterns while maintaining reconstruction ability without being too fine-grained for tractable transition modeling.

### 4.1. Coarse and Fine Codebook Training

To address this, we adopt a residual codebook learning strategy with two distinct codebooks: a *coarse codebook* $\mathcal{C}_c = \{\mathbf{e}_c\}_{c=1}^{n_c}$ with a limited number of codes $n_c$ to capture primary temporal patterns, and a *fine codebook* $\mathcal{C}_f = \{\mathbf{e}_f\}_{f=1}^{n_f}$ with a larger capacity ($n_c \ll n_f$) to capture residual details, where $\mathbf{e}_c, \mathbf{e}_f \in \mathbb{R}^{d_{\text{dim}}}$. A codebook is a finite set of learnable vectors that serve as reference points for quantizing continuous latent representations into discrete codes through euclidean distance assignments. The quantization process for a single patch token $\mathbf{z} \in \mathbb{R}^{d_{\text{dim}}}$ follows:

$$\tilde{c} = \arg\min_c \|\ell_2(\mathbf{z}) - \ell_2(\mathbf{e}_c)\|_2^2; \text{ where } \mathbf{e}_c \in \mathcal{C}_c$$
$$\tilde{f} = \arg\min_f \|\ell_2(\mathbf{z}) - \ell_2(\mathbf{e}_{\tilde{c}}) - \ell_2(\mathbf{e}_f)\|_2^2; \text{ where } \mathbf{e}_f \in \mathcal{C}_f$$

where $\mathbf{z}$ is first mapped to a coarse code and the residual $\ell_2(\mathbf{z}) - \ell_2(\mathbf{e}_c)$ is quantized with a fine code. $\tilde{c}$ and $\tilde{f}$ denote the selected coarse and fine codebook indices, respectively, and $\ell_2$ is the L2 normalization which converts the Euclidean distance into cosine similarity assignment (Yu et al., 2021). After quantization, we perform regular VQVAE (Van Den Oord et al., 2017) training, where we optimize the VQ loss $\mathcal{L}_{\text{VQ}} = \mathcal{L}_{\text{code}} + \mathcal{L}_{\text{rec}}$. The $\mathcal{L}_{\text{code}}$ optimizes the codebook and $\mathcal{L}_{\text{rec}}$ is used with a decoder $\mathcal{D}_\phi$ (*i.e.* transformer) to reconstruct the original time series $\mathbf{X}$.

$$\mathcal{L}_{\text{code}} = \|\operatorname{sg}[\mathbf{z}] - \mathbf{e}_c\|_2^2 + \beta \|\operatorname{sg}[\mathbf{e}_c] - \mathbf{z}\|_2^2$$
$$+ \|\operatorname{sg}[\mathbf{z} - \mathbf{e}_c] - \mathbf{e}_f\|_2^2 + \beta \|\operatorname{sg}[\mathbf{e}_f] - (\mathbf{z} - \mathbf{e}_c)\|_2^2$$
$$\mathcal{L}_{\text{rec}} = \|\mathbf{X} - \mathcal{D}_\phi(\mathbf{e}_c + \mathbf{e}_f)\|_2^2$$

$\beta$ is a commitment term which we set to 0.25 and sg is a stop gradient operation. We also train a classifier head using the $[\text{CLS}] \in \mathbb{R}^{d_{dim}}$ token that has been encoded using $\mathcal{E}_\theta$, optimizing the cross-entropy loss $\mathcal{L}_{\text{ce}}$ as the classification objective. The final loss function for source training is

$$\mathcal{L}_{\text{src}} = \mathcal{L}_{\text{ce}} + \mathcal{L}_{\text{VQ}}. \tag{1}$$

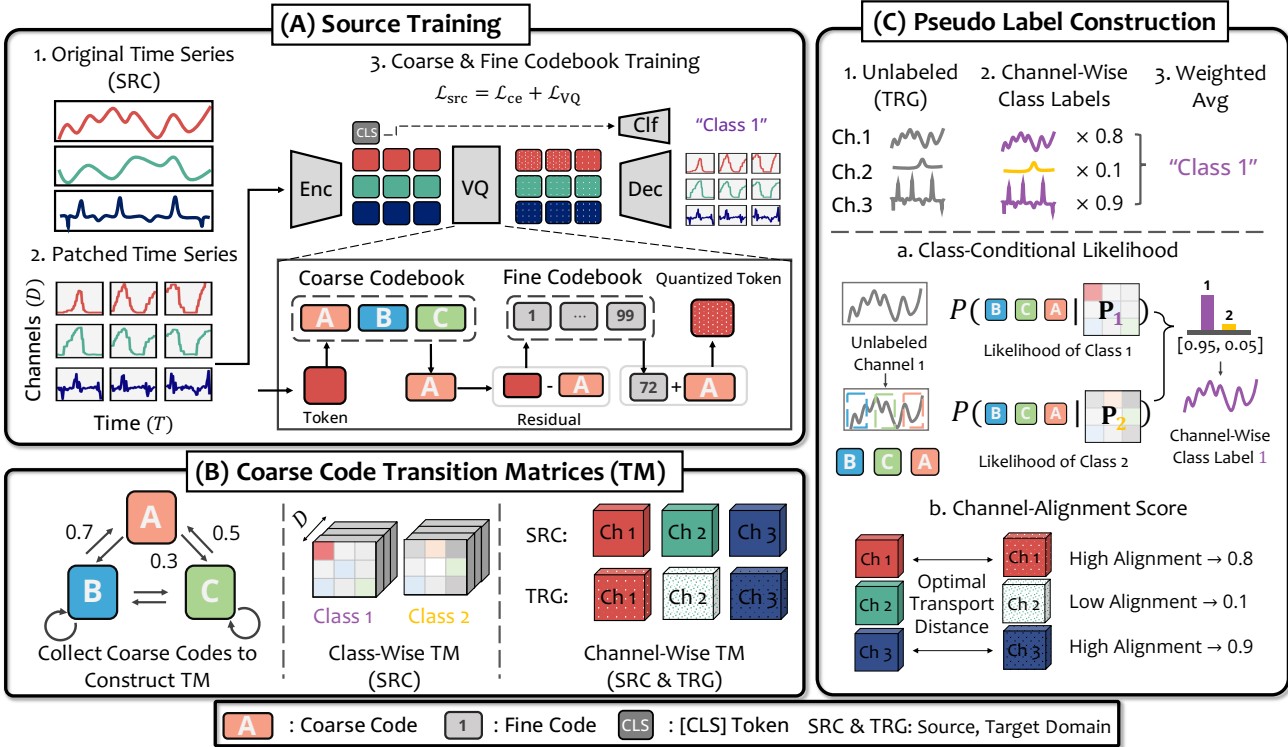

Figure 1: **Overall scheme of TransPL.** **(A) Source Training**: We first train the whole model architecture, *i.e.*, encoder, decoder, two VQ codebooks (coarse and fine codebook), and classifier, using the labeled source domain data. A `[CLS]` token is appended to the input patches and is used as the input to the classifier. **(B) Coarse Code Transition Matrices (TM)**: The trained encoder and coarse codebook infer coarse codes from both target (unlabeled) and source (labeled) domains. These codes serve as states for constructing class-wise TM (from source) and channel-wise TM (from both domains). **(C) Pseudo Label Construction**: For unlabeled target data, we compute class-conditional likelihoods per channel using class-wise TM to obtain channel-wise class labels. These are weighted by the similarity between source and target channel-wise TMs, then averaged to generate final pseudo labels.

**Design method.** The hierarchical structure of mapping to limited coarse codes followed by residual fine codes draws inspiration from classical time series additive decomposition methods (Brockwell & Davis, 2002), where a time series can be decomposed into trend, seasonal, and residual. For simplicity, we adopt a two-level representation where the coarse encodes the short-term trend of the patches and the fine maps the residual details. The coarse code can be thought of as capturing the essential time dynamics and is used for coarse code transition matrices, while the fine code is solely used for stable VQVAE training. To validate the roles of the coarse and fine codes, we analyze their permutation entropy (PE), a measure of complexity (Bandt & Pompe, 2002). Figure 2 shows that the reconstructed coarse code has lower PE compared to fine codes in various time series tasks, aligning with our expectation that the coarse code captures the global trends, while the fine code encodes the residual details. Table 1 compares different codebook size combinations, showing that our design method maintains strong reconstruction performance while having zero dead codes (*i.e.* codes that are not being used), enabling tractable computation for transition matrices.

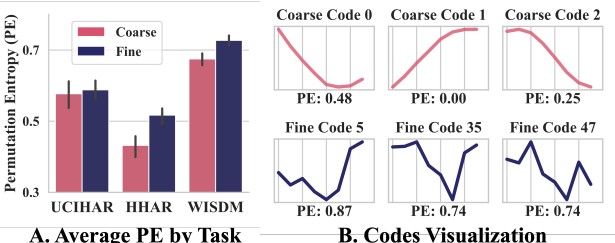

Figure 2: **A. Average PE**: The average permutation entropy (PE) score for the reconstructed coarse and fine VQ codes. PE is a measure of temporal complexity (Bandt & Pompe, 2002), where a lower PE indicates a simpler pattern. Across all time series benchmarks (*i.e.* UCIHAR, HHAR, WISDM), the coarse code exhibits lower PE compared to the fine code. **B. Visualization**: The reconstructed coarse and fine VQ codes shows that the coarse code has simplistic patterns (*e.g.* upward, downward trends), while the fine code has more complex patterns. Full results in Appendix B.

### 4.2. Coarse Code Transition Matrices (TMs)

After source training, we freeze all learnable parameters and infer the coarse codes from the training sets of both the source and target domains. We now view each coarse code sequence as a discrete Markov chain, where the codes

**Table 1:** Comparison of single VQVAE (top three rows) versus our residual coarse-fine VQ codebook configurations (bottom rows) on HHAR task. Results are averages of 10 source-target pairs.

| Coarse | Fine | MSE ↓ | PL Acc ↑ | PL MF1 ↑ | Dead Code (%) ↓ |
|--------|------|-------|----------|----------|------------------|
| 8   | -  | 0.214 | 65.4 | 63.8 | 0.1 |
| 64  | -  | 0.115 | 58.8 | 55.7 | 22.5 |
| 128 | -  | 0.106 | 59.4 | 54.9 | 66.8 |
| 64  | 64 | **0.077** | 58.6 | 54.7 | 20.9 |
| 64  | 8  | 0.093 | 59.1 | 54.9 | 22.2 |
| 8   | 8  | 0.126 | 65.6 | 63.6 | **0.0** |
| 8   | 64 | 0.090 | **68.4** | **66.9** | **0.0** |

represent the states of the chain. Let $s_t$ denote the state at time step $t = [1, \ldots, N]$, and $\mathcal{C}_c = \{\mathbf{e}_c\}_{c=1}^{n_c}$ be the set of $n_c$ possible coarse codes (states). The transition probabilities between states are estimated based on the observed code sequences in the training data, under the assumption that the probability of transitioning to a particular state depends only on the current state (*i.e.* 1-step Markov property). Formally, the transition probability from state $i$ to state $j$ is given by:

$$p(s_{t+1} = \mathbf{e}_j | s_t = \mathbf{e}_i) = \frac{\text{count}(\mathbf{e}_i, \mathbf{e}_j)}{\text{count}(\mathbf{e}_i)} \qquad (2)$$

where $\text{count}(\mathbf{e}_i, \mathbf{e}_j)$ is the number of occurrences of the transition from code $\mathbf{e}_i$ to $\mathbf{e}_j$ in the training sequences, and $\text{count}(\mathbf{e}_i)$ is the total number of occurrences of code $\mathbf{e}_i$.

Using the inferred coarse codes, we construct three sets of coarse code transition matrices (TM): class-wise TM from the source domain $\mathbf{P}_{\text{cl}}^{\mathcal{S}}$, and channel-wise $\mathbf{P}_{\text{ch}}^{\mathcal{S}}$ and $\mathbf{P}_{\text{ch}}^{\mathcal{T}}$ from the source and target domains, respectively. Note that we can only construct the class-wise TM for the source, as we do not have access to the label information of the target.

**Class-Wise TM (Source).** Based on the covariate shift assumption $\mathcal{P}(y|\mathbf{X})^{\mathcal{S}} = \mathcal{P}(y|\mathbf{X})^{\mathcal{T}}$, the temporal sequences from the same class should have similar coarse code transition patterns between source and target, while sequences from different classes should have distinct transition patterns. As such, from the labeled source training data, we collect the coarse code sequences and construct the TM for *each class and channel*, leading to $\mathbf{P}_{\text{cl}}^{\mathcal{S}} \in \mathbb{R}^{K \times D \times n_c \times n_c}$, where $K$ is the number of classes, $D$ is number of channels, and $n_c$ is the number of coarse codes. Each class-wise TM $\mathbf{P}_{\text{cl},k}^{\mathcal{S}}$, where $k \in \{1, \ldots, K\}$, is used to calculate the channel-wise class-conditional likelihood $p(\mathbf{X}^d|y = k)$ in Bayes' theorem for constructing pseudo-labels in the target domain. Calculating the class-conditional likelihood is similar to how we perform the maximum likelihood estimation in Hidden Markov Models (Bishop & Nasrabadi, 2006). This likelihood measures the probability of observing a given coarse code sequence from channel $d$ of $\mathbf{X}$ conditioned on the class label $y = k$, based on the transition patterns observed from the source domain.

**Channel-Wise TM (Source & Target).** Since $\mathcal{P}_X^{\mathcal{S}} \neq \mathcal{P}_X^{\mathcal{T}}$, the marginal of the input data differs between the source

and target. As such, we construct $\mathbf{P}_{\text{ch}}^{\mathcal{S}}$, $\mathbf{P}_{\text{ch}}^{\mathcal{T}} \in \mathbb{R}^{D \times n_c \times n_c}$ for *each channel* $d = \{1, \ldots, D\}$ by constructing the TM in a channel-wise manner. Unlike the class-wise TM $\mathbf{P}_{\text{cl}}^{\mathcal{S}}$, the $\mathbf{P}_{\text{ch}}^{\mathcal{S}}$ and $\mathbf{P}_{\text{ch}}^{\mathcal{T}}$ are constructed without considering the class labels. The channel-wise TM captures the transition patterns specific to each channel in both domains, later used for channel-wise alignment calculation in Section 5.

## 5. Target Adaptation

### 5.1. Pseudo Labeling

**Strategy.** We propose a pseudo-labeling approach that leverages transition matrices (TM) for domain adaptation. Given an unlabeled time series instance $\mathbf{X}$ from the target, we construct a pseudo label vector $\hat{\mathbf{y}} = [\hat{y}_1, \ldots, \hat{y}_k]^{\top}$, where $\hat{y}_k$ is the logits for class $k$ and is calculated by:

$$\hat{y}_k = \frac{1}{D} \sum_{d=1}^{D} w_d \; \frac{p(\mathbf{X}^d|y = k) \, p(k)}{\sum_{c=1}^{K} p(\mathbf{X}^d|y = c) \, p(c)} \qquad (3)$$

The pseudo label can be seen as a *weighted aggregation of channel-wise class posteriors*, where the posterior is formulated using Bayes' theorem (Bishop & Nasrabadi, 2006). The $p(\mathbf{X}^d|y = k)$ is the class conditional likelihood of observing the coarse code sequence in channel $d$ given the class-wise TM $\mathbf{P}_{\text{cl},k}^{\mathcal{S}}$ and $p(k)$ is the prior label distribution of label $k$ in the target domain. The channel-wise class posteriors are weighted by the channel alignment score $w_d$ for all channels $d = [1, \ldots, D]$. Intuitively, class posteriors from channels that are shifted less are weighted more.

**Channel Alignment via Optimal Transport.** When a channel shift occurs between the source and target domains, the change is expected to manifest in the coarse code transition patterns, which can be captured by the channel-wise TMs. For notational simplicity, let $\mathbf{p}_{(i,:)}^{\mathcal{S}}, \mathbf{p}_{(i,:)}^{\mathcal{T}} \in \mathbb{R}^{n_c}$ represent the transition probability vectors from code $i$ to all other codes $j = [1, \ldots, n_c]$ in channel $d$ of the source and target domains, respectively, with $\sum_{j=1}^{n_c} \mathbf{p}_{(i,j)}^{\mathcal{S}} = \sum_{j=1}^{n_c} \mathbf{p}_{(i,j)}^{\mathcal{T}} = 1$. In practice, we add a small constant $\epsilon$ to each element and re-normalize the rows to handle numerical instabilities. Here, simply measuring the difference between the two vectors based on measures such as the Euclidean distance does not take into account the code-wise semantics. For instance, coarse codes A and B might capture similar patterns. As such, the transitions A→A and A→B should be considered semantically similar, even though they correspond to different indices in the transition probability vectors.

To address this, we use optimal transport (Peyré et al., 2019) to measure the shift between the source and target transition probabilities. Optimal transport finds a transportation plan $\gamma_i^* \in \mathbb{R}_+^{n_c \times n_c}$ that minimizes the cost of transporting $\mathbf{p}_{(i,:)}^{\mathcal{S}}$ into $\mathbf{p}_{(i,:)}^{\mathcal{T}}$, where the cost is defined by a cost matrix $\mathbf{M} \in \mathbb{R}_+^{n_c \times n_c}$ that encodes the semantic distance be-

tween the coarse codes. Formally, the optimal transport problem for code $i$ is defined as:

$$\gamma_i^* = \underset{\gamma_i \in \mathbb{R}_+^{n_c \times n_c}}{\arg\min} \langle \gamma_i, \mathbf{M} \rangle \quad \text{s.t. } \gamma_i \mathbf{1} = \mathbf{p}_{(i,:)}^{\mathcal{S}}, \ \gamma_i^\top \mathbf{1} = \mathbf{p}_{(i,:)}^{\mathcal{T}}$$
(4)

where $\langle \cdot, \cdot \rangle$ denotes the Frobenius dot product, and $\mathbf{1}$ is an all-ones vector. The constraints ensure that the marginal distributions of the transportation plan match the source and target transition probabilities. The cost matrix $\mathbf{M}_{(i,j)}$ is the cosine distance between the coarse codes:

$$\mathbf{M}_{(i,j)} = 1 - \mathbf{e}_i^\top \cdot \mathbf{e}_j / \|\mathbf{e}_i\| \|\mathbf{e}_j\|.$$
(5)

The intuition behind the cost matrix is that the cost is low when the transition happens between similar codes. This aligns with the idea that transitioning between semantically similar codes should be less costly than transitioning between dissimilar codes. Once the transportation plan $\gamma_i^*$ is obtained, we can compute the channel alignment score $w_d$:

$$w_d = \exp\left(-\left(\frac{1}{n_c}\sum_{i=1}^{n_c} \langle \gamma_i^*, \mathbf{M} \rangle\right)^2 / \sigma^2\right)$$
(6)

where $\langle \gamma_i^*, \mathbf{M} \rangle$ is the Earth mover's distance (Pele & Werman, 2009) between the source and target probability vectors. The $\exp(\cdot)$ is the RBF kernel with $\sigma$ as the bandwidth hyperparameter, effectively converting the distance to an alignment score. The channel alignment score has an intuitive meaning: place higher weights for channels that have shifted less, while placing smaller weights for channels that have undergone strong domain shift. This weighting scheme prioritizes class labels from more reliable channels.

**Class Conditional Likelihood** The $p(\mathbf{X}^d | y = k)$ is the class conditional likelihood of observing coarse code sequence of $\mathbf{X}^d$ given class-wise TM for class $k$ (Section 4.2). Assuming a first-order Markov property, we calculate the likelihood of the sequence as the product of the individual transition probabilities $\prod_{t=1}^{N-1} p(s_{t+1} | s_t, y = k)$. Here, $s_t$ and $s_{t+1}$ are the coarse codes at steps $t$ and $t+1$ in the sequence $\mathbf{X}^d$. In practice, to ensure numerical stability, we compute the log-likelihood instead:

$$\log p(\mathbf{X}^d | y = k) = \frac{1}{N} \sum_{t=1}^{N-1} \log p(s_{t+1} | s_t, y = k) \quad (7)$$

**Incorporating Prior.** In UDA, the model does not have access to the label of the target distribution. As such, we set $p(k)$, the prior label distribution of class $k$ to a uniform distribution. However, the distribution of the label might be accessible as a form of weak supervision (*e.g.* self-report on the proportion of time a subject has spent on each exercise in HAR) and has been known as domain adaptation with *weak supervision* (Wilson et al., 2020). Following this, we can naturally incorporate the label distribution as prior $p(k)$ in

Bayes' theorem when such information is given. Following the log computation in class conditional likelihood, the log prior is used: $\log p(k) / \tau$ where $\tau$ is a temperature hyperparameter that modulates the strength of the prior. Searching for an appropriate $\tau$ was important in our analysis, as too strong $\tau$ leads to degraded model performance by forcing the model to predict the dominant class.

**Putting it all together.** We compute the pseudo label for the target domain instance by: (1) computing the channel-wise class posteriors based on the likelihoods and label priors, and (2) weighting the channel-wise class posteriors based on the channel alignment scores that capture sensor shifts.

### 5.2. Pseudo Label Training With Target Domain

After constructing the pseudo label vector $\hat{\mathbf{y}}$ for the target domain's training set, we use the $\arg\max_k \hat{\mathbf{y}}$ as the label. For each mini-batch during training, we select the top $r_{\text{top}}$ percent of samples based on their confidence scores, defined as $\max_k \hat{\mathbf{y}}_k$, to form a reliable subset for model updates. We fine-tune all model parameters from the source model (*i.e.* encoder, decoder, classifier, and codebooks) and optimize the following objective similar to $\mathcal{L}_{\text{src}}$ in Equation (1):

$$\mathcal{L}_{\text{trg}} = \lambda_1 \mathcal{L}_{\text{ce}} + \lambda_2 \mathcal{L}_{\text{VQ}}.$$
(8)

The $\lambda_1$ and $\lambda_2$ are the weighting coefficients, which can be replaced with learnable parameters adopted from multi-task learning (Kendall et al., 2018) to avoid extensive search.

## 6. Experiments

We introduce the datasets, experiment setups, and comparison baselines in our work. The details are in Appendix C.

**Datasets.** We consider four time series datasets: UCI-HAR (Anguita et al., 2013), WISDM (Kwapisz et al., 2011), HHAR (Stisen et al., 2015), and PTB (Bousseljot et al., 1995). The first three tasks are human activity recognition tasks where we set each user as a single domain and select 10 pairs of source and target domains. The source-target pairs are exactly the same as the benchmark suite performed by AdaTime (Ragab et al., 2023). PTB is an electrocardiogram (ECG) dataset, where we select each age group as domains and assess the adaptation performance between four different pairs of different age groups.

**Experiments.** We used the labeled source and unlabeled target's training set for training, and the labeled source domain's test set as the validation set. We evaluated the performance of the adapted model on the target domain's test set (source risk setup (Ragab et al., 2023)). We utilized the same patch encoder from the works of (Gui et al., 2024) and re-implemented all baselines using the same encoder. We report the average accuracy and macro-F1 score (MF1) over the different domains, with full results in Appendix D.

**Table 2: Adaptation Performance in Target.** Mean accuracy (Acc) and macro F1 (MF1) over 10 pairs. Best in **bold**, second best underlined. Full results in Appendix D.

| Dataset | UCIHAR | | WISDM | | HHAR | | PTB | |
|---|---|---|---|---|---|---|---|---|
| | Acc | MF1 | Acc | MF1 | Acc | MF1 | Acc | MF1 |
| No Adapt | 57.0 | 53.6 | 59.8 | 49.1 | 55.7 | 52.0 | 50.5 | 56.9 |
| DeepCoral | 62.0 | 57.6 | 61.9 | 52.0 | 57.6 | 54.9 | 57.4 | 67.8 |
| MMDA | 60.8 | 54.0 | 60.1 | 52.0 | 58.7 | 55.6 | 60.0 | 70.1 |
| CoDATS | 62.7 | 58.5 | 63.7 | 48.3 | 61.4 | 59.4 | 56.1 | 67.6 |
| SASA | 57.2 | 51.4 | 60.9 | 45.6 | 60.3 | 56.5 | 62.4 | 73.8 |
| RAINCOAT | 58.6 | 61.1 | 47.3 | 63.0 | 58.6 | 61.1 | 59.8 | 41.2 |
| SoftMax | 62.4 | 57.4 | 61.8 | 52.7 | 62.5 | 59.4 | 62.3 | 70.7 |
| NCP | 59.4 | 53.9 | 56.7 | 50.0 | 51.5 | 45.6 | 60.6 | 67.7 |
| SP | 62.6 | 59.2 | 51.4 | 45.8 | 50.8 | 47.4 | 60.6 | 67.2 |
| ATT | 56.2 | 46.1 | 58.7 | 47.5 | 59.2 | 51.8 | 55.7 | 59.8 |
| SHOT | 67.8 | 64.3 | 62.2 | 54.6 | 64.8 | 63.2 | 61.6 | 66.9 |
| T2PL | 63.8 | 60.6 | 57.3 | 53.3 | 64.1 | 62.8 | 59.0 | 67.0 |
| TransPL | **69.0** | **64.9** | **64.0** | 56.2 | **68.4** | **65.3** | **67.2** | **74.0** |

**Baselines.** We compared TransPL with several DA methods such as DeepCoral (Sun & Saenko, 2016), MMDA (Rahman et al., 2020), CoDATS (Wilson et al., 2020), SASA (Cai et al., 2021), RAINCOAT (He et al., 2023a), and pseudo labeling strategies such as Softmax (Lee et al., 2013), NCP (Wang & Breckon, 2020), SP (Wang & Breckon, 2020), ATT (Saito et al., 2017), SHOT (Liang et al., 2020), and T2PL (Liu et al., 2023a).

**Implementation.** We selected the number of coarse and fine codes to $n_c = 8$, $n_f = 64$ for all tasks. The patch length was set to $m = 15$ for PTB, and the rest to $m = 8$. The detailed configurations and search ranges for TransPL and baselines are provided in Appendix E.

# 7. Results

## 7.1. Performance on UDA

In Table 2, we report the mean accuracy and mean macro F1 (MF1) of the target domain's test set across all baselines and datasets, with the full results in Appendix D. Following the evaluation protocol of AdaTime (Ragab et al., 2023), we maintain strict experimental consistency by utilizing identical source-target pairs (10 pairs for HAR tasks, four pairs for PTB) without any selective sampling. TransPL demonstrates the best adaptation performance across all four datasets, achieving large improvements over the non-adapted baseline with average gains of 11.4% in accuracy and 12.2% in MF1. The performance gain was obtained by explicitly modeling the temporal transitions and incorporating channel-wise shifts. Notably, when compared to the strongest baseline SHOT, TransPL maintains a consistent advantage of an average 3.0% in accuracy and 2.9% in MF1. These results are significant as they are achieved using a uniform label prior distribution; as demonstrated in Section 7.3, incorporating the true class distribution prior can lead to more substantial performance gains.

## 7.2. Accuracy of Pseudo Labels

We report the accuracy of the constructed pseudo label of TransPL in Table 3 over the unlabeled target domain's training set, alongside other pseudo labeling methods. In practice, the pseudo label's performance is not used or known during model training as we do not have access to the true labels of the target domain. However, this analysis provides valuable insights into the quality of our pseudo labels and their contribution to the overall adaptation performance. We first demonstrate that TransPL is the best pseudo labeling method compared to existing pseudo label strategies such as SoftMax, NCP, SP, ATT, SHOT, and T2PL. The average performance gains are 6.1% and 4.9% compared to the best results of all baselines. In addition, using the true label distribution for the prior $p(k)$ as a weak supervision (WS) shows that the gains widen to 10.7% and 5.2% for accuracy and MF1, respectively. Also, we note that using WS leads to enhanced pseudo labeling performance in seven out of eight cases for TransPL, showcasing that our method effectively leverages prior knowledge when available while maintaining robust performance even without such information.

**Table 3: Pseudo-labeling accuracy in target training.** Weak supervision (WS) denotes the use of prior label information.

| Dataset | UCIHAR | | WISDM | | HHAR | | PTB | |
|---|---|---|---|---|---|---|---|---|
| | Acc | MF1 | Acc | MF1 | Acc | MF1 | Acc | MF1 |
| SoftMax | 63.8 | 59.6 | 60.9 | 51.9 | 60.3 | 57.2 | 68.2 | 78.7 |
| NCP | 57.9 | 54.0 | 52.2 | 45.8 | 47.3 | 42.2 | 62.7 | 71.6 |
| SP | 57.4 | 54.5 | 49.9 | 46.2 | 48.0 | 46.6 | 59.8 | 68.5 |
| ATT | 55.7 | 45.7 | 58.8 | 46.5 | 57.8 | 50.4 | 58.9 | 64.0 |
| SHOT | 66.1 | 64.3 | 57.7 | 53.3 | 62.2 | 61.2 | 65.5 | 75.1 |
| T2PL | 62.9 | 60.8 | 56.3 | 53.5 | 61.3 | 60.7 | 65.5 | 75.1 |
| 1D-CNN | 65.8 | 62.5 | 50.2 | 39.8 | 51.5 | 47.3 | 68.1 | 48.2 |
| LSTM | 62.2 | 58.6 | 42.3 | 29.1 | 46.8 | 42.8 | 64.4 | 46.9 |
| GRU | 61.3 | 56.5 | 43.8 | 33.3 | 48.9 | 44.6 | 65.4 | 47.6 |
| TransPL | **71.0** | **67.8** | 61.8 | **56.1** | 68.4 | 66.9 | 80.4 | 86.7 |
| TransPL (+WS) | 74.2 | 68.7 | 69.3 | 53.0 | 69.0 | 67.4 | 87.7 | 89.3 |

**Use of Transition Matrix.** TransPL is a novel pseudo labeling approach, where the joint distribution $\mathcal{P}(\mathbf{X}, y)^{\mathcal{S}}$ of the source is modeled through the coarse code transition matrices (TM). The use of TM to model the joint distribution of time series is beneficial for several reasons. First, the use of TM aggregates transition patterns across sequences, providing robustness against temporal variations and noise common in time series data. Second, TM enables the explicit modeling of class-conditional patterns and channel-wise shifts that are unique characteristics in time series adaptation. To demonstrate, we compared our generative approach to the direct modeling of the coarse codes using discriminative approaches using models such as 1D-CNN, LSTM (Hochreiter, 1997), and GRU (Cho et al., 2014). Utilizing the same data as in TransPL, these sequential models show significantly sub-optimal pseudo labeling performance as in Table 3, failing to capture domain-invariant temporal dynamics present in the source domain.

## 7.3. Weakly Supervised UDA

We compared TransPL with CoDATS (Wilson et al., 2020) under the assumption that the true class label distribution of target training is known as a form of weak supervision (WS). For instance, in human activity recognition, while individual sample labels are difficult to obtain, users can self-report their time allocation across different activities. CoDATS incorporates WS during model training by minimizing the Kullback-Leibler (KL) divergence between the true label distribution and the expected target class predictions in each mini-batch. In contrast, TransPL leverages this distribution during the pseudo-labeling stage, employing it as a prior $p(k)$ in the Bayes formulation described in Equation (3).

**Table 4: Comparing weak supervision (WS) performance** between CoDATS and TransPL.

| Dataset | Metric | CoDATS | | TransPL | |
|---------|--------|--------|------------|---------|------------|
|         |        | Base | +WS (Gain) | Base | +WS (Gain) |
| UCIHAR | Acc | 62.7 | 62.3 (-0.4) | 69.0 | 71.2 (+2.2) |
|        | MF1 | 58.5 | 58.4 (-0.1) | 64.9 | 67.1 (+2.2) |
| WISDM | Acc | 63.7 | 67.1 (+3.4) | 64.0 | 71.3 (+7.3) |
|       | MF1 | 48.3 | 50.8 (+2.6) | 56.2 | 57.9 (+1.6) |
| HHAR | Acc | 61.4 | 60.8 (-0.6) | 68.4 | 70.4 (+2.0) |
|      | MF1 | 59.4 | 59.1 (-0.3) | 65.3 | 67.3 (+2.0) |
| PTB | Acc | 56.1 | 49.8 (-5.3) | 67.2 | 72.4 (+5.2) |
|     | MF1 | 67.6 | 60.0 (-6.5) | 74.0 | 77.3 (+3.3) |

In Table 4, we show that utilizing WS is beneficial for TransPL in all datasets, while CoDATS shows mixed results. Notably, CoDATS exhibits performance degradation in UCIHAR, HHAR, and PTB tasks, suggesting that weak supervision can potentially hinder model performance by introducing prediction bias. In contrast, TransPL achieves consistent performance improvements in both accuracy and MF1 across all tasks. These results highlight TransPL's robustness even in challenging scenarios where source and target domains exhibit different class distributions (Ragab et al., 2023), as observed in WISDM and PTB.

## 7.4. Class Conditional Likelihoods

The class-wise transition matrices (TMs) derived from TransPL provide interpretable insights into the robustness and discriminative power of our pseudo-labeling process. Figure 3 illustrates this through the analysis of class-conditional likelihoods for an unlabeled target sequence. Here, TransPL successfully identifies the true class (class 2) by assigning it the highest likelihood, while also revealing meaningful similarities with class 0 through comparable temporal patterns. The zero likelihood assigned to class 4 further demonstrates the method's discriminative ability, as this class exhibits a distinctly different temporal pattern with static transitions between identical coarse codes. Most importantly, the TMs capture these temporal relationships despite significant amplitude variations between source and

target sequences, highlighting our method's invariance to amplitude shifts while preserving temporal dynamics. As such, unlike black-box classifiers or clustering-based approaches, TransPL's pseudo labeling process provides transparent insights into the pseudo-labeling process.

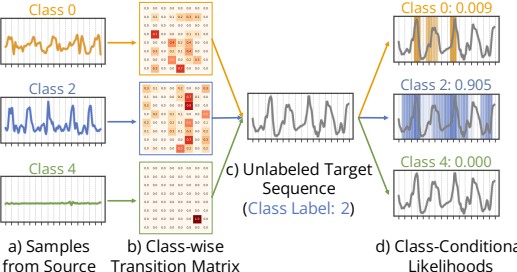

**Figure 3: Class conditional likelihood visualization in UCI.** Samples from the source are used to construct the class-wise transition matrices (TMs). Then, these TMs are used to calculate the class-conditional likelihoods of the unlabeled target sequence.

## 7.5. Ablation Study

We conducted ablation experiments in Table 5 to evaluate the joint usage of channel alignment (CA) weighting and prior label distribution as weak supervision (WS) in TransPL. The results demonstrate that both modules contribute to improved domain adaptation performance, whether applied individually or in combination. In seven out of eight cases, the best results were obtained when both modules were used together. Here, the performance gain from the use of CA indicates that it is beneficial to model the channel-wise shifts in multivariate time series for domain adaptation. Additionally, utilizing prior label distribution as WS enhances the model's adaptation capability. The findings confirm that CA and WS address complementary aspects of time series adaptation, making their joint usage beneficial.

**Table 5:** Ablation of channel alignment (CA) and weak supervision (WS) modules in the adaptation performance.

| CA | WS | UCIHAR | | WISDM | | HHAR | | PTB | |
|----|----|--------|------|-------|------|------|------|------|------|
|    |    | Acc | MF1 | Acc | MF1 | Acc | MF1 | Acc | MF1 |
| ✗ | ✗ | 68.0 | 63.5 | 62.3 | 51.0 | 63.2 | 59.5 | 68.3 | 73.3 |
| ✓ | ✗ | 69.0 | 64.9 | 64.0 | 56.2 | 68.4 | 65.3 | 67.2 | 74.0 |
| ✗ | ✓ | 68.6 | 63.8 | 69.2 | 55.2 | 62.3 | 58.8 | 72.2 | 77.9 |
| ✓ | ✓ | 71.2 | 67.1 | 71.3 | 57.9 | 70.4 | 67.3 | 72.4 | 77.3 |

## 7.6. Channel Alignment Analysis

We analyzed the effectiveness of our proposed channel alignment (CA) distance measure by examining its representation space and its ability to capture source-target channel-wise shifts. Figure 4 presents a channel-wise visualization of source and target samples, revealing varying degrees of domain shift across channels. Notably, channel three exhibits a more pronounced distributional shift between the source and target domains compared to channels one and two. By com-

paring our distance measure to clustering-based approaches such as prototype distance, we show that prototype-based methods fail to capture the channel-wise shifts, as they aggregate features across all samples and ignore the temporal dynamics inherent in time series. However, our CA module effectively captures the shift from the source to target by computing the average costs of transporting each code transition vector between the source and target, demonstrating its ability to address channel shifts in time series.

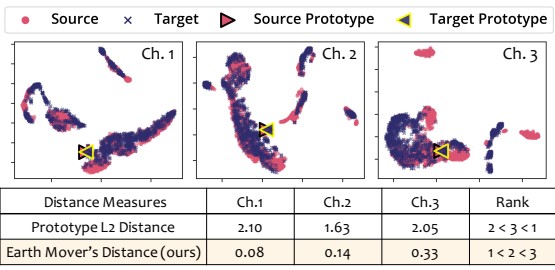

| Distance Measures | Ch.1 | Ch.2 | Ch.3 | Rank |
|---|---|---|---|---|
| Prototype L2 Distance | 2.10 | 1.63 | 2.05 | 2 < 3 < 1 |
| Earth Mover's Distance (ours) | 0.08 | 0.14 | 0.33 | 1 < 2 < 3 |

**Figure 4: Channel alignment (CA) analysis.** We visualized the latent representation of each channels's `[CLS]` token and observe that the degree of shift differs between channels. Specifically, channel 3 shows higher degrees of shift between source and target compared to channel 1 and 2. We show that using prototype distances (as in clustering based approaches) lead to inaccurate distance measurement, while our earth mover's distance from the CA module provides well calibrated distance between channels.

## 8. Limitation

We acknowledge the following limitations of our work. First, our work assigns lower weights to channels that are heavily shifted during the adaptation phase. However, only certain channels may contain class-relevant information (Kim et al., 2024), and when class information is concentrated in channels experiencing significant domain shift, our method would assign lower weights to these channels, leading to degraded adaptation performance. Future works can incorporate channel importance measures alongside ours, allowing for weighting that considers both channel shift and the importance of the channel for classification. Second, while we empirically observed that the coarse codes capture more general patterns and the fine codes capture more fine-grained details, this is not regularized mathematically. We may consider adding regularization terms to enforce such hierarchical relationships explicitly.

## 9. Conclusion

We propose TransPL, a novel pseudo labeling method for time series domain adaptation, which incorporates coarse and fine vector quantized codes to model the temporal transitions through VQ-code transition matrices (TMs). The TMs enable the explicit modeling of temporal transitions and channel-wise shifts in time series, enabling us to im-

prove adaptation performance. Moreover, the transparent nature of our pseudo labeling provides interpretable insights at each stage, making it both powerful and explainable.

## Acknowledgment

This work was supported by the Institute of Information & communications Technology Planning & Evaluation(IITP) grant funded by the Korea government(MSIT) (No.RS-2024-00508465) and Institute of Information & communications Technology Planning & Evaluation(IITP) grant funded by the Korea government(MSIT) (No.RS-2020-II201336, Artificial Intelligence Graduate School Program(UNIST)).

The authors acknowledge Dr. Seokju Hahn and Dr. Yoontae Hwang for their insightful discussions that significantly contributed to the development of this work. We also thank the anonymous reviewers for providing insightful feedback and going through our manuscript.

## Impact Statement

This paper presents work whose goal is to advance the field of unsupervised domain adaptation of time series. Previous works have mostly focused on simply enhancing the adaptation performance but have not provided explainable insights into "what" is being shifted in time series. Here, our TransPL specifically addresses this problem by modeling the temporal transitions and channel-wise shifts in multivariate time series through the use of coarse code transition matrices. We strongly believe that our approach of using the coarse code patterns provides novel insights into the time series community.

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

## A. Notations.

| Symbol | Description | Dimension |
|---|---|---|
| $\mathbf{X}$ | Original time series instance | $\mathbb{R}^{D \times T}$ |
| $\mathbf{X}^{\text{re}}$ | Original time series instance reshaped into patches | $\mathbb{R}^{D \times N \times m}$ |
| $y$ | True label of time series instance | $\mathbb{R}$ |
| $\mathcal{S}$ | Source | - |
| $\mathcal{T}$ | Target | - |
| $\mathcal{P}_X, \mathcal{P}_Y$ | Time series instance and label distributions | - |
| $D$ | Number of channels (sensors) | - |
| $T$ | Time sequence length | - |
| $m$ | Size of patch | - |
| $N$ | Total number of patch in a single time series sequence | - |
| $d_{\text{dim}}$ | Dimension of latent representation | - |
| $n_c, n_f$ | Number of coarse and fine codes | $8, 64$ |
| $\mathcal{C}_c$ | Coarse codebook | $\mathbb{R}^{d_{\text{dim}} \times n_c}$ |
| $\mathcal{C}_f$ | Fine codebook | $\mathbb{R}^{d_{\text{dim}} \times n_f}$ |
| $\mathbf{e}_c, \mathbf{e}_f$ | Coarse and fine code vector | $\mathbb{R}^{d_{\text{dim}}}$ |
| $\mathbf{P}^{\mathcal{S}}_{\text{cl}}$ | Class-wise coarse code transition matrix from source | $\mathbb{R}^{K \times D \times n_c \times n_c}$ |
| $\mathbf{P}^{\mathcal{S}}_{\text{ch}}$ | Channel-wise coarse code transition matrix from source | $\mathbb{R}^{D \times n_c \times n_c}$ |
| $\mathbf{P}^{\mathcal{T}}_{\text{ch}}$ | Channel-wise coarse code transition matrix from target | $\mathbb{R}^{D \times n_c \times n_c}$ |
| $\mathbf{M}$ | Cost matrix | $\mathbb{R}^{n_c \times n_c}_{+}$ |

**Table 6:** Notation and Symbols

# B. Coarse and Fine Codes

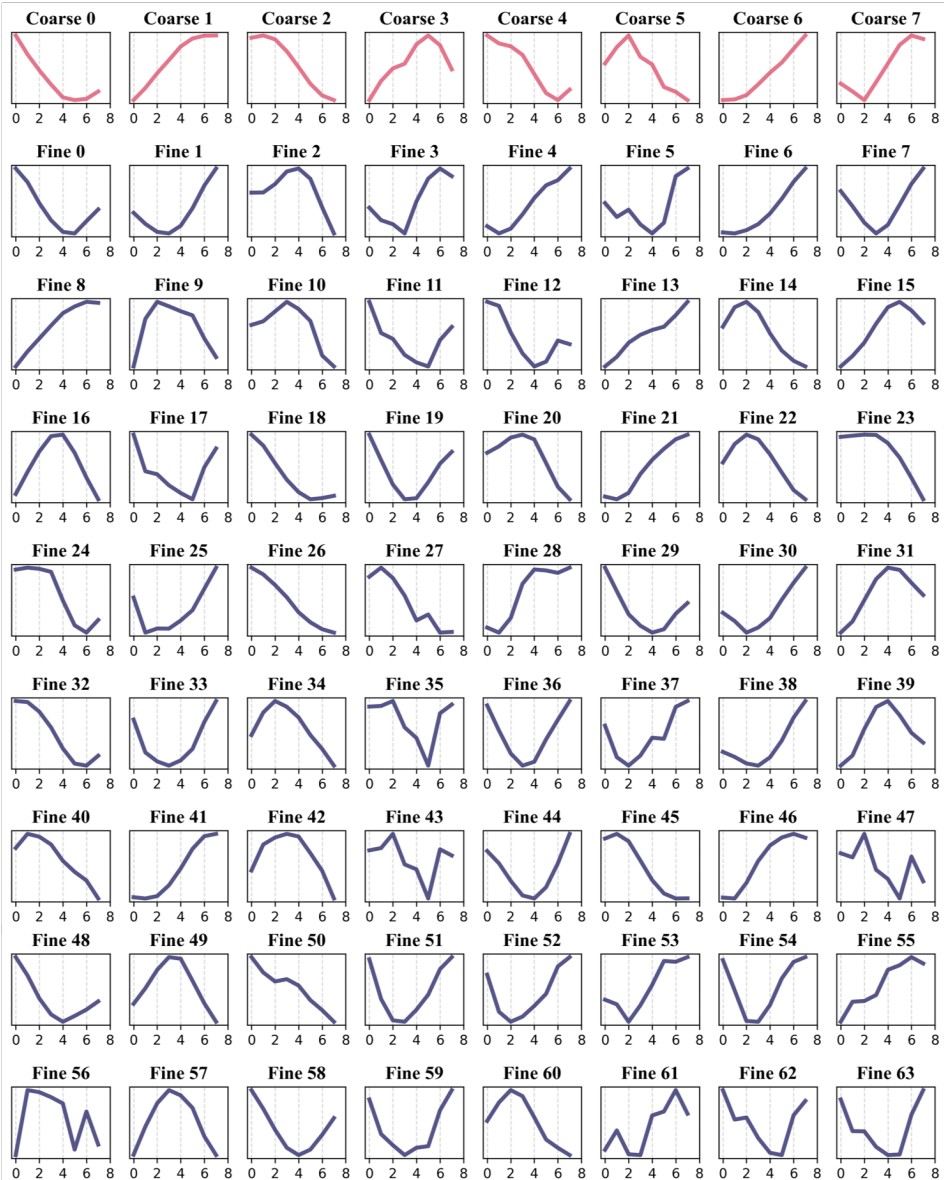

**Figure 5: Visualization of the full coarse and fine codes in HHAR task.** We visualized the trained coarse and fine codes from the HHAR task $2 \rightarrow 11$. The top row contains the 8 coarse codes, and the rest are 64 fine codes. We observe that the fine codes are capturing more fine grained detailed compared to the coarse codes.

# C. Experiment Details

**Hardware and Software.** We performed all experiments on a single NVIDIA RTX A6000-48GB GPU. We used Python 3.10 and PyTorch 2.0.1. For the main experiment pipelines, we used PyTorch Lightning 2.1.2. All baselines were re-implemented within our frameworks.

**Datasets.** We used four datasets to test our domain adaptation algorithm: UCIHAR, WISDM, HHAR, and PTB. The UCIHAR, WISDM, HHAR tasks are human activity recognition (HAR) datasets and are well-known benchmark datasets for domain adaptation in time series. We used the same datasets and splits provided in AdaTime (Ragab et al., 2023). PTB is an ECG dataset, where the task is to classify five different classes of diseases. However, a more processed binary classification (Myocardial infarction vs Healthy Control) dataset was suggested by (Wang et al., 2024), and we used this dataset. The detailed distribution of classes and setup of domains are shown in Table 7.

**Table 7: Distribution of PTB Task.** We did not used (N.U) age groups 20-30, 70-80, and 80-90 as there were not enough balanced class samples. We set age groups 30-40 as Domain 1, 40-50 as Domain 2, 50-60 as Domain 3, and 60-70 as Domain 4.

| Age Group | Label | Patient Count | Domain |
|---|---|---|---|
| 20-30 | Healthy control | 13 | N.U |
| 30-40 | Healthy control
Myocardial infarction | 11
4 | 1 |
| 40-50 | Healthy control
Myocardial infarction | 6
25 | 2 |
| 50-60 | Healthy control
Myocardial infarction | 9
43 | 3 |
| 60-70 | Healthy control
Myocardial infarction | 6
47 | 4 |
| 70-80 | Myocardial infarction | 23 | N.U |
| 80-90 | Healthy control
Myocardial infarction | 1
5 | N.U |

**Source and Domain Pairs.** For UCIHAR, WISDM, and HHAR, we used the exact same ten different splits provided in AdaTime (Ragab et al., 2023). For PTB, we utilized four different pairs of source and target, as shown below.

- UCIHAR: $2 \to 11, 6 \to 23, 7 \to 13, 9 \to 18, 12 \to 16, 18 \to 27, 20 \to 5, 24 \to 8, 28 \to 27, 30 \to 20$.

- WISDM: $7 \to 18, 20 \to 30, 35 \to 31, 17 \to 23, 6 \to 19, 2 \to 11, 33 \to 12, 5 \to 26, 28 \to 4, 23 \to 32$.

- HHAR: $0 \to 6, 1 \to 6, 2 \to 7, 3 \to 8, 4 \to 5, 5 \to 0, 6 \to 1, 7 \to 4, 8 \to 3, 0 \to 2$.

- PTB: $1 \to 3, 1 \to 4, 3 \to 4, 4 \to 1$

| Dataset | Task Type | Domains | Length | # Channels | # Class |
|---|---|---|---|---|---|
| UCIHAR | HAR | 30 | 128 | 9 | 6 |
| WISDM | HAR | 36 | 128 | 3 | 6 |
| HHAR | HAR | 9 | 128 | 3 | 6 |
| PTB | ECG | 4 | 300 | 15 | 2 |

**Table 8:** Details of the datasets used in our main experiments

# D. Full Results

**Table 9: UCIHAR Results Acc.** Results across different source-target domain pairs.

| Algorithm | 2→11 | 6→23 | 7→13 | 9→18 | 12→16 | 18→27 | 20→5 | 24→8 | 28→27 | 30→20 | Average |
|---|---|---|---|---|---|---|---|---|---|---|---|
| No Adapt | 60.0 | 60.7 | 79.8 | 40.0 | 60.9 | 65.5 | 48.4 | 58.8 | 47.8 | 47.7 | 57.0 |
| DeepCoral | 63.2 | 78.8 | 67.3 | 59.3 | 61.1 | 75.2 | 59.3 | 60.0 | 61.1 | 52.3 | 62.0 |
| MMDA | 65.3 | 77.8 | 57.3 | 36.3 | 61.9 | 77.0 | 36.3 | 61.2 | 61.9 | 59.8 | 60.8 |
| CoDATS | 55.8 | 77.8 | 71.8 | 35.2 | 59.3 | 78.8 | 35.2 | 63.5 | 59.3 | 63.6 | 62.7 |
| SASA | 61.1 | 80.8 | 57.3 | 36.3 | 58.4 | 78.8 | 36.3 | 54.1 | 58.4 | 57.0 | 57.2 |
| RAINCOAT | 34.6 | 65.3 | 49.5 | 75.1 | 62.4 | 37.5 | 75.1 | 80.0 | 62.4 | 58.0 | 58.6 |
| SoftMax | 65.3 | 76.8 | 68.2 | 56.0 | 67.3 | 69.0 | 56.0 | 62.4 | 67.3 | 39.3 | 62.4 |
| NCP | 48.4 | 81.8 | 68.2 | 36.3 | 67.3 | 71.7 | 36.3 | 65.9 | 67.3 | 44.9 | 59.4 |
| SP | 56.8 | 80.8 | 63.6 | 51.6 | 64.6 | 62.8 | 51.6 | 71.8 | 64.6 | 50.5 | 62.6 |
| ATT | 80.0 | 70.7 | 51.8 | 30.8 | 49.6 | 81.4 | 30.8 | 56.5 | 49.6 | 53.3 | 56.2 |
| SHOT | 65.3 | 80.8 | 75.5 | 59.3 | 90.3 | 75.2 | 59.3 | 64.7 | 90.3 | 43.0 | 67.8 |
| T2PL | 65.3 | 72.7 | 72.7 | 57.1 | 69.9 | 63.7 | 57.1 | 69.4 | 69.9 | 36.4 | 63.8 |
| TransPL | 75.8 | 84.8 | 67.3 | 59.3 | 66.4 | 89.4 | 59.3 | 75.3 | 66.4 | 61.7 | **69.0** |

**Table 10: UCIHAR Results F1.** Results across different source-target domain pairs.

| Algorithm | 2→11 | 6→23 | 7→13 | 9→18 | 12→16 | 18→27 | 20→5 | 24→8 | 28→27 | 30→20 | Average |
|---|---|---|---|---|---|---|---|---|---|---|---|
| No Adapt | 52.7 | 54.3 | 77.6 | 38.2 | 56.0 | 64.9 | 48.0 | 57.1 | 49.3 | 37.5 | 53.6 |
| DeepCoral | 55.7 | 55.5 | 76.1 | 35.1 | 65.7 | 72.2 | 57.4 | 55.6 | 59.1 | 43.8 | 57.6 |
| MMDA | 55.7 | 61.9 | 71.0 | 35.6 | 52.9 | 73.0 | 29.9 | 52.8 | 57.9 | 49.1 | 54.0 |
| CoDATS | 46.7 | 60.1 | 70.8 | 54.7 | 73.3 | 76.0 | 32.2 | 55.4 | 61.7 | 61.7 | 58.5 |
| SASA | 52.4 | 63.2 | 77.3 | 17.0 | 48.0 | 76.3 | 32.2 | 49.4 | 49.8 | 48.2 | 51.4 |
| RAINCOAT | 34.9 | 63.5 | 68.7 | 60.6 | 57.3 | 44.6 | 75.2 | 81.9 | 65.3 | 59.1 | 61.1 |
| SoftMax | 56.2 | 70.8 | 74.0 | 39.7 | 65.8 | 64.5 | 56.7 | 53.2 | 63.9 | 28.7 | 57.4 |
| NCP | 35.3 | 44.6 | 79.0 | 43.7 | 69.9 | 66.5 | 38.4 | 63.2 | 64.7 | 33.4 | 53.9 |
| SP | 48.5 | 62.2 | 78.4 | 51.3 | 60.4 | 59.3 | 50.6 | 72.9 | 60.4 | 48.7 | 59.2 |
| ATT | 77.4 | 36.4 | 61.3 | 28.5 | 41.0 | 71.4 | 23.6 | 46.9 | 39.4 | 35.2 | 46.1 |
| SHOT | 56.5 | 69.7 | 77.9 | 48.7 | 77.5 | 70.4 | 58.1 | 62.7 | 87.7 | 34.2 | 64.3 |
| T2PL | 57.0 | 71.8 | 70.9 | 51.1 | 74.9 | 59.5 | 53.5 | 69.5 | 67.1 | 30.5 | 60.6 |
| TransPL | 70.4 | 63.1 | 82.3 | 35.9 | 64.6 | 88.1 | 56.2 | 72.7 | 63.5 | 52.6 | **64.9** |

**Table 11: WISDM Results Acc.** Results across different source-target domain pairs.

| Algorithm | 7→18 | 20→30 | 35→31 | 17→23 | 6→19 | 2→11 | 33→12 | 5→26 | 28→4 | 23→32 | Average |
|---|---|---|---|---|---|---|---|---|---|---|---|
| No Adapt | 80.2 | 64.1 | 66.3 | 48.3 | 62.1 | 31.6 | 60.9 | 73.2 | 83.3 | 27.5 | 59.8 |
| DeepCoral | 83.0 | 57.3 | 66.3 | 53.3 | 56.8 | 50.0 | 65.5 | 74.4 | 83.3 | 29.0 | 61.9 |
| MMDA | 70.8 | 19.4 | 59.0 | 71.7 | 75.8 | 42.1 | 73.6 | 73.2 | 54.5 | 60.9 | 60.1 |
| CoDATS | 74.5 | 80.6 | 54.2 | 70.0 | 54.5 | 47.4 | 82.8 | 75.6 | 78.8 | 18.8 | 63.7 |
| SASA | 67.9 | 62.1 | 67.5 | 58.3 | 59.8 | 47.4 | 77.0 | 75.6 | 80.3 | 13.0 | 60.9 |
| RAINCOAT | 53.1 | 58.5 | 40.2 | 30.6 | 58.9 | 77.8 | 37.0 | 37.8 | 67.6 | 11.8 | 47.3 |
| SoftMax | 76.4 | 52.4 | 67.5 | 60.0 | 51.5 | 46.1 | 74.7 | 74.4 | 83.3 | 31.9 | 61.8 |
| NCP | 67.9 | 49.5 | 67.5 | 46.7 | 56.8 | 59.2 | 59.8 | 47.6 | 83.3 | 29.0 | 56.7 |
| SP | 55.7 | 44.7 | 59.0 | 35.0 | 55.3 | 61.8 | 48.3 | 46.3 | 84.8 | 23.2 | 51.4 |
| ATT | 82.1 | 49.5 | 49.4 | 75.0 | 39.4 | 44.7 | 71.3 | 73.2 | 87.9 | 14.5 | 58.7 |
| SHOT | 71.7 | 58.3 | 67.5 | 41.7 | 78.8 | 46.1 | 47.1 | 68.3 | 83.3 | 59.4 | 62.2 |
| T2PL | 67.9 | 53.4 | 69.9 | 45.0 | 79.5 | 46.1 | 41.4 | 37.8 | 78.8 | 53.6 | 57.3 |
| Ours | 84.9 | 66.0 | 63.9 | 66.7 | 48.5 | 61.8 | 88.5 | 74.4 | 54.5 | 30.4 | **64.0** |

**Table 12: WISDM Results F1.** Results across different source-target domain pairs.

| Algorithm | 7→18 | 20→30 | 35→31 | 17→23 | 6→19 | 2→11 | 33→12 | 5→26 | 28→4 | 23→32 | Average |
|---|---|---|---|---|---|---|---|---|---|---|---|
| No Adapt | 65.9 | 55.5 | 56.1 | 35.5 | 58.5 | 48.3 | 38.5 | 39.2 | 74.5 | 19.3 | 49.1 |
| DeepCoral | 66.3 | 50.9 | 56.5 | 32.5 | 47.8 | 63.3 | 56.3 | 39.7 | 75.5 | 31.5 | 52.0 |
| MMDA | 67.1 | 26.1 | 54.3 | 47.2 | 69.5 | 51.4 | 61.4 | 39.3 | 62.5 | 41.2 | 52.0 |
| CoDATS | 50.3 | 73.3 | 27.2 | 39.0 | 42.5 | 54.7 | 71.6 | 41.4 | 68.7 | 14.0 | 48.3 |
| SASA | 32.4 | 47.3 | 61.1 | 35.4 | 50.6 | 38.9 | 67.2 | 41.5 | 73.6 | 7.7 | 45.6 |
| RAINCOAT | 63.5 | 78.1 | 60.2 | 67.7 | 75.9 | 75.8 | 48.8 | 71.1 | 68.8 | 19.8 | **63.0** |
| SoftMax | 66.5 | 47.5 | 60.4 | 40.1 | 45.2 | 59.4 | 59.4 | 39.7 | 76.8 | 32.2 | 52.7 |
| NCP | 43.8 | 51.0 | 59.4 | 24.6 | 51.3 | 65.7 | 56.1 | 31.6 | 83.6 | 32.6 | 50.0 |
| SP | 54.3 | 44.2 | 36.8 | 16.7 | 51.5 | 62.4 | 45.9 | 31.4 | 78.8 | 36.1 | 45.8 |
| ATT | 67.8 | 38.9 | 44.9 | 47.4 | 31.1 | 51.5 | 56.7 | 39.5 | 78.8 | 18.2 | 47.5 |
| SHOT | 67.8 | 56.3 | 59.6 | 21.5 | 75.5 | 57.5 | 44.8 | 37.6 | 79.4 | 46.1 | 54.6 |
| T2PL | 65.8 | 52.2 | 63.7 | 31.5 | 72.6 | 57.3 | 43.6 | 20.8 | 82.5 | 43.4 | 53.3 |
| Ours | 68.9 | 66.1 | 58.3 | 44.9 | 39.8 | 63.3 | 80.9 | 39.6 | 64.3 | 36.5 | 56.2 |

**Table 13: HHAR Results Acc.** Results across different source-target domain pairs.

| Algorithm | 0→6 | 1→6 | 2→7 | 3→8 | 4→5 | 5→0 | 6→1 | 7→4 | 8→3 | 0→2 | Average |
|---|---|---|---|---|---|---|---|---|---|---|---|
| No Adapt | 37.3 | 56.9 | 45.1 | 63.0 | 63.1 | 47.5 | 73.3 | 63.5 | 55.4 | 51.8 | 55.7 |
| DeepCoral | 37.7 | 56.1 | 54.5 | 63.2 | 65.0 | 35.2 | 73.3 | 70.7 | 69.8 | 50.7 | 57.6 |
| MMDA | 38.3 | 55.3 | 57.2 | 53.0 | 57.3 | 47.3 | 80.2 | 76.0 | 61.1 | 61.3 | 58.7 |
| CoDATS | 41.7 | 64.3 | 62.0 | 75.8 | 63.2 | 35.9 | 59.1 | 80.0 | 72.9 | 58.7 | 61.4 |
| SASA | 44.1 | 52.9 | 57.6 | 69.0 | 71.8 | 37.9 | 72.8 | 63.7 | 70.5 | 62.5 | 60.3 |
| RAINCOAT | 34.6 | 62.6 | 65.3 | 61.1 | 49.5 | 37.5 | 75.1 | 80.0 | 62.4 | 58.0 | 58.6 |
| SoftMax | 38.3 | 51.9 | 59.9 | 70.2 | 76.8 | 45.7 | 79.7 | 67.9 | 73.1 | 61.3 | 62.5 |
| NCP | 35.7 | 46.1 | 59.1 | 42.1 | 51.8 | 54.5 | 52.2 | 60.9 | 57.1 | 55.2 | 51.5 |
| SP | 30.5 | 44.7 | 58.5 | 44.8 | 57.4 | 48.1 | 47.2 | 55.5 | 59.7 | 61.5 | 50.8 |
| ATT | 36.3 | 52.1 | 51.6 | 77.4 | 68.3 | 47.5 | 63.6 | 76.0 | 59.5 | 59.6 | 59.2 |
| SHOT | 35.9 | 47.5 | 58.2 | 77.6 | 86.5 | 44.0 | 81.7 | 73.9 | 70.7 | 71.6 | 64.8 |
| T2PL | 37.9 | 64.7 | 55.3 | 73.3 | 89.4 | 37.9 | 75.2 | 77.4 | 63.7 | 66.5 | 64.1 |
| Ours | 39.5 | 73.1 | 60.5 | 72.7 | 75.4 | 52.3 | 77.1 | 89.2 | 80.3 | 64.2 | **68.4** |

**Table 14: HHAR Results F1.** Results across different source-target domain pairs.

| Algorithm | 0→6 | 1→6 | 2→7 | 3→8 | 4→5 | 5→0 | 6→1 | 7→4 | 8→3 | 0→2 | Average |
|---|---|---|---|---|---|---|---|---|---|---|---|
| No Adapt | 34.0 | 49.2 | 40.1 | 63.4 | 57.4 | 38.7 | 72.8 | 60.9 | 56.3 | 47.3 | 52.0 |
| DeepCoral | 33.0 | 50.7 | 50.2 | 66.3 | 58.5 | 30.0 | 71.8 | 69.5 | 71.3 | 47.9 | 54.9 |
| MMDA | 35.9 | 48.2 | 51.1 | 54.7 | 50.8 | 36.4 | 80.8 | 75.1 | 63.0 | 60.1 | 55.6 |
| CoDATS | 39.6 | 59.7 | 60.5 | 76.2 | 57.1 | 32.3 | 58.1 | 79.5 | 56.5 | 56.5 | 59.4 |
| SASA | 41.4 | 45.7 | 52.5 | 68.6 | 64.1 | 28.7 | 71.4 | 61.8 | 69.8 | 60.9 | 56.5 |
| RAINCOAT | 34.9 | 63.5 | 68.7 | 60.6 | 57.3 | 44.6 | 75.2 | 81.9 | 65.3 | 59.1 | 61.1 |
| SoftMax | 37.3 | 45.0 | 53.3 | 72.0 | 70.3 | 37.5 | 78.5 | 66.4 | 74.3 | 59.2 | 59.4 |
| NCP | 29.6 | 39.1 | 48.6 | 34.5 | 46.2 | 44.3 | 46.3 | 59.5 | 57.1 | 50.4 | 45.6 |
| SP | 29.8 | 40.6 | 56.6 | 36.3 | 52.2 | 40.8 | 42.9 | 54.1 | 61.8 | 59.2 | 47.4 |
| ATT | 27.5 | 43.2 | 40.4 | 74.8 | 57.4 | 34.5 | 54.8 | 72.2 | 56.4 | 56.4 | 51.8 |
| SHOT | 35.1 | 45.6 | 53.7 | 76.2 | 86.9 | 37.7 | 80.3 | 72.9 | 72.7 | 70.7 | 63.2 |
| T2PL | 37.3 | 59.1 | 51.7 | 73.8 | 89.9 | 33.8 | 74.6 | 76.3 | 66.7 | 64.9 | 62.8 |
| Ours | 37.3 | 72.5 | 53.9 | 72.1 | 68.1 | 43.0 | 74.5 | 89.5 | 80.4 | 61.8 | **65.3** |

**Table 15: PTB Results Acc.** Results across different source-target domain pairs.

| Algorithm | 1→3 | 1→4 | 3→4 | 4→1 | Average |
|---|---|---|---|---|---|
| No Adapt | 25.7 | 35.4 | 92.3 | 48.4 | 50.5 |
| DeepCoral | 47.5 | 55.2 | 89.7 | 37.4 | 57.4 |
| MMDA | 39.7 | 74.0 | 92.2 | 33.8 | 60.0 |
| CoDATS | 40.6 | 57.7 | 92.3 | 33.8 | 56.1 |
| SASA | 58.4 | 65.1 | 92.3 | 33.8 | 62.4 |
| RAINCOAT | 45.6 | 53.1 | 95.6 | 45.0 | 59.8 |
| SoftMax | 45.4 | 65.4 | 92.0 | 46.5 | 62.3 |
| NCP | 36.5 | 58.6 | 79.0 | 68.5 | 60.6 |
| SP | 36.4 | 58.6 | 74.8 | 72.8 | 60.6 |
| ATT | 86.7 | 7.8 | 91.7 | 36.6 | 55.7 |
| SHOT | 37.6 | 59.6 | 84.6 | 64.4 | 61.6 |
| T2PL | 35.5 | 59.0 | 87.8 | 53.9 | 59.0 |
| Ours | 51.9 | 72.7 | 94.4 | 49.7 | **67.2** |

**Table 16: PTB Results MF1.** Results across different source-target domain pairs.

| Algorithm | 1→3 | 1→4 | 3→4 | 4→1 | Average |
|---|---|---|---|---|---|
| No Adapt | 32.6 | 46.4 | 96.0 | 52.6 | 56.9 |
| DeepCoral | 59.2 | 68.5 | 94.5 | 48.8 | 67.8 |
| MMDA | 49.7 | 84.2 | 96.0 | 50.5 | 70.1 |
| CoDATS | 53.1 | 70.9 | 96.0 | 50.5 | 67.6 |
| SASA | 71.3 | 77.4 | 96.0 | 50.5 | 73.8 |
| RAINCOAT | 39.3 | 40.4 | 54.1 | 31.0 | 41.2 |
| SoftMax | 58.3 | 77.6 | 95.8 | 50.9 | 70.7 |
| NCP | 47.2 | 71.9 | 87.8 | 64.0 | 67.7 |
| SP | 47.1 | 71.9 | 84.9 | 64.8 | 67.2 |
| ATT | 92.9 | 0.1 | 95.6 | 50.4 | 59.8 |
| SHOT | 48.8 | 72.8 | 91.5 | 54.6 | 66.9 |
| T2PL | 46.1 | 72.3 | 93.5 | 56.2 | 67.0 |
| Ours | 65.0 | 83.3 | 97.0 | 50.8 | **74.0** |

# E. Baseslines and TransPL Setup

## E.1. Baselines

For all Baselines, we used the official implementations or used the implementations from AdaTime (Ragab et al., 2023). For a fair comparison, we utilized the same patch transformer encoder structure as in our work. The batch size was set to 32 for all works. We also conducted a hyperparameter grid search for the learning rates between $[0.001, 0.002, 0.0002, 0.0005]$, reporting the learning rates with the best MF1 scores. Moreover, certain methods employ multiple loss functions for adaptation. As in our work, we employed a learnable parameter to automatically adjust the weights between these losses.

## E.2. TransPL

To implement TransPL, we used the patch transformer encoder used in (Gui et al., 2024) to encode time series patches. The patch representations were then quantized using our coarse and fine codebooks. The coarse was first mapped to the patch representation, where the residuals were mapped to the fine codebooks. Here, we used $n_c = 8$ and $n_f = 64$ numbers of codes for coarse and fine codebooks, respectively. The transition matrices are constructed using only the coarse codes, where the transitions are calculated using a vectorized approach. We also employed the K-Means initialization for the codebooks using the first mini-batch samples in the source domain. We utilized the same encoder architecture for the decoder of VQVAE.

| Task | $d_{\text{dim}}$ | Batch | Max Epoch | Patch Length ($m$) | $\sigma$ | $\tau$ | $r_{\text{top}}$ | Source LR | Adaptation LR |
|---|---|---|---|---|---|---|---|---|---|
| UCIHAR | 64 | 32 | 200 | 8 | 0.2 | 1.0 | 0.5 | 0.0005 | 0.0005 |
| WISDM | 128 | 32 | 200 | 8 | 0.1 | 2.0 | 0.2 | 0.0002 | 0.0005 |
| HHAR | 128 | 32 | 200 | 8 | 0.1 | 5.0 | 0.2 | 0.0002 | 0.0002 |
| PTB | 128 | 32 | 200 | 15 | 0.2 | 1.0 | 0.7 | 0.0005 | 0.0005 |

Table 17: TransPL Hyperparameter Configurations.

# F. Different Patch Length

Table 18: Performance of Different Patch Lengths

| | UCIHAR | | WISDM | | HHAR | |
|---|---|---|---|---|---|---|
| Length | Acc | MF1 | Acc | MF1 | Acc | MF1 |
| $m = 4$ | 69.0 | 64.5 | 64.2 | 52.6 | 64.7 | 60.7 |
| $m = 8$ | 69.0 | 64.9 | 64.0 | 56.2 | 68.4 | 65.3 |
| $m = 16$ | 65.4 | 60.7 | 64.7 | 56.6 | 64.5 | 60.0 |
| $m = 32$ | 67.1 | 63.2 | 56.5 | 49.1 | 55.0 | 51.5 |

Table 19: Performance of Different Patch Lengths in PTB

| | PTB | |
|---|---|---|
| Length | Acc | MF1 |
| $m = 5$ | 69.9 | 75.5 |
| $m = 10$ | 63.2 | 68.1 |
| $m = 15$ | 67.2 | 74.0 |
| $m = 30$ | 68.7 | 76.5 |

## G. Channel-level corruption

To further analyze our channel-wise alignment strategy, we conducted channel-level corruption experiments on the UCIHAR task across all ten source-target pairs. As TransPL assumes to place lower weights on channels that have shifted strongly, we expect that our algorithm should place lower weights on channels where noise is added. From the UCIHAR task with ten source-target pairs, we identified that the sixth channel (starting from zero index) consistently exhibited high values across most of these pairs. When we introduced increasing levels of noise (Gaussian) to this specific channel, we observed a corresponding decrease in values, confirming the utility of our proposed channel-level adaptation.

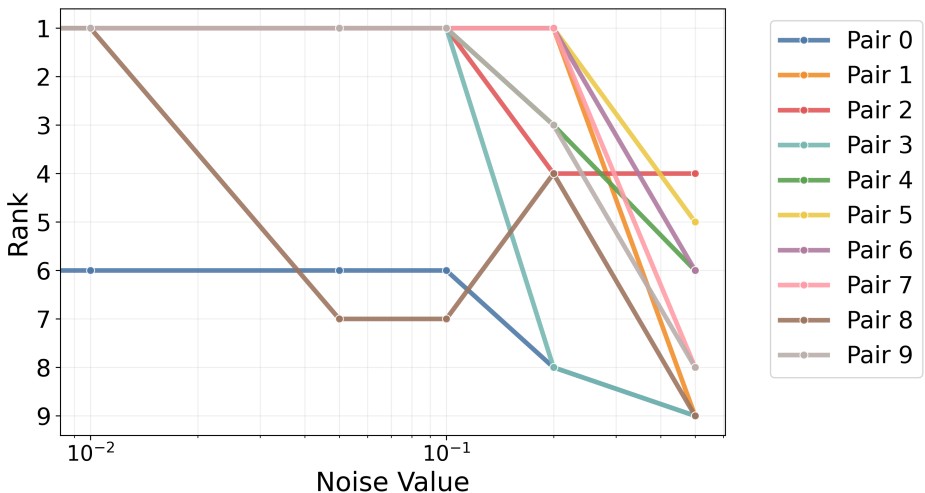

**Figure 6: Visualization of the ranks of $w_6$ for the UCIHAR task.** Our results demonstrate that as noise magnitude increases in the sixth channel, its rank consistently decreases across all tested pairs, confirming that our channel alignment module successfully identifies channels that have gone through channel shift.

