# OpenReview forum: "TransPL: VQ-Code Transition Matrices for Pseudo-Labeling of Time Series Unsupervised Domain Adaptation"
_ICML.cc/2025/Conference — ICML 2025 poster_

### Official Review · Reviewer_GrxU · 2025-03-13

**Overall Recommendation:** 2

**Summary:**

The paper presents TransPL, a novel unsupervised domain adaptation (UDA) method for time series data that improves pseudo-labeling by capturing temporal transitions and channel-wise shifts between domains. Traditional pseudo-labeling methods fail to model these patterns, leading to suboptimal labels. TransPL addresses this by constructing class- and channel-wise code transition matrices using vector quantization (VQ) of time series patches from the source domain and applying Bayes’ rule for adaptation to the target domain.

**Claims And Evidence:**

The author claims that the Permutation Entropy of UCIHAR, HHAR, and WISDM data sets can prove that fine VQ code is more complex than coarse VQ code. This can indeed be seen from the visual representation.

**Essential References Not Discussed:**

It is recommended to join ACON[1] in the future.

[1] Liu, M., Chen, X., Shu, Y., Li, X., Guan, W., & Nie, L. (2024). Boosting Transferability and Discriminability for Time Series Domain Adaptation. Advances in Neural Information Processing Systems, 37, 100402-100427.

**Experimental Designs Or Analyses:**

Yes, I found the experimental results to be unreasonable. (listed on questions)

**Methods And Evaluation Criteria:**

The experimental design and benchmark dataset are nearly consistent with the protocols of previous studies in this field.

**Other Comments Or Suggestions:**

1. Typos:
Line 171: “quantization”

2. As mentioned above, since this method does not use both source and target domain data simultaneously, it may be more aligned with the field of SFUDA.

**Other Strengths And Weaknesses:**

Strengths:
(1) The paper introduces a creative adaptation of vector quantization (VQ) techniques to the UDA setting.
(2) The integration of Bayesian inference with learned transition matrices offers a unique way to improve label reliability.

Weaknesses: As questions

**Questions For Authors:**

1.	The term weakly-supervised UDA is somewhat confusing. Please clarify its meaning (including in the abstract) to avoid misunderstandings. If this terminology has been used in prior research, please provide proper citations.
2.	This pipeline does not necessarily require VQVAE and could potentially perform better with a Mixture of Experts approach. Could you explain why VQVAE is essential to this framework?
3.	Since VQVAE heavily relies on representation learning quality, the performance of the reconstruction task is a key factor influencing the domain adaptation task. Although Table 1 presents the MSE loss, could you provide further insights into the relationship between the reconstruction task and domain adaptation performance?
4.	How can we justify that the representations learned for the classification task and the reconstruction task are similar rather than different? This could significantly impact how VQVAE learns representations. Given that this manuscript emphasizes interpretability, we would appreciate further clarification on this point.
5.	It is unlikely that every dataset would achieve Zero Dead Code under the same codebook size. Could you explain how Zero Dead Code is ensured in this approach?
6.	Why did you choose Earth Mover’s Distance?
7.	Since this manuscript follows the AdaTime evaluation protocol, which is the same as ACON [1], why do other methods (e.g., RAINCOAT, CoDATS, DeepCoral) perform significantly worse under the same dataset and scenario? Please provide a thorough explanation; otherwise, this discrepancy raises concerns about the validity of the manuscript’s findings.

**Relation To Broader Scientific Literature:**

Since this method does not use both source and target domain data simultaneously, it may be more aligned with the field of “source-free unsupervised domain adaptation” (SFUDA).

**Theoretical Claims:**

This manuscript only uses the very basic Bayes’ rule.

---

> ### Author Rebuttal · Authors · 2025-03-30
>
> We thank the reviewer for thoroughly going through our manuscript, and providing valuable comments. We want to highlight points that were misunderstood and should be clarified. While we wish to thank you individually for each point, we are constrained by the length limit. Thank you for your understanding.
>
> **C1. Missing Ref.** ACON. We will add them to our updated manuscript.
>
> **C2. SFUDA.** This is a misunderstanding. Our TransPL is not source-free, as it utilizes the source training data alongside the pseudo-labeled target training data for adaptation, as noted in Line 162: *where the model is fine-tuned using the labeled source and pseudo-labeled target data for domain adaptation.* We note that our work can be extended to the SFUDA, if we only utilize the target data for adaptation, but we believe this is out of our scope where our current focus is on the UDA setup. However, based on the reviewer's insight, we found that this could be better clarified. As such, we added *The adaptation is complete by fine-tuning the model with both labeled source and the pseudo-labeled target training set* in Lines 321.
>
> **Q1. Weakly-Sup. UDA.** The term Weakly-Supervised means that the label distribution of the target domain is given as an additional source of information. Such scenarios are highly practical as users might not label all of their data but can self-report on the proportion of time they have spent on each exercise. This term was first used in time series by CoDATS [KDD'20] (we would like to address the reviewer to Lines 100-109 and Lines 353-355, where we have provided explanations and proper citations needed).
>
> **Q2. VQVAE.** The use of VQVAE is essential, and all the proposed works build on top of VQ. We are not aware of other architectures that could better replace VQVAE. Simply, a patch can now be represented as discrete codes. These discrete codes are then used for modeling the temporal transitions and channel-wise shifts, which are the main contributions of this work. Here, VQ enables the **mapping of each time series patch** into coarse code vectors. This makes the semantics of the time series intact, while making the transition matrices sensible.
>
> **Q3. Recon. Performance.** The reviewer is correct. Reconstruction is important for our use, as good reconstruction ensures meaningful code vectors to be constructed, which is critical for valid transition matrices. Constructing non-representative code vectors that can't represent the time series task at hand leads to meaningless transition matrices, leading to degraded pseudo label accuracy. In Table 1, we show that obtaining good reconstruction performance as well as making the transition meaningful (by limiting the # of coarse codes) are both essential to making accurate pseudo labels.
>
> **Q4. Recon. and Classification.** While there is no strict guarantee that the representations for classification and reconstruction are identical, we argue that they are well aligned for the following:
>
> 1. **Empirical Evidence**: Our results show strong performance in both classification and reconstruction across multiple data, with no signs of overfitting to one task. The pseudo labels consistently achieve the best results, and both losses are well optimized, leading to stable outcomes.
> 2. **Prior Works**: Prior works in time series [1,2] have successfully used VQVAE for both reconstruction and classification. Additionally, optimizing auxiliary tasks alongside reconstruction is a common practice in the broader literature [3].
>
> [1] TOTEM: TOkenized Time Series EMbeddings for General TS [TMLR'24]
>
> [2] VQ Pretraining for EEG TS with Random Projection and Phase Alignment [ICML'24]
>
> [3] DeWave: Discrete EEG Waves Encoding for Brain Dynamics [NeurIPS '23]
>
> Q5. While we have empirically observed that zero dead coarse code was achieved in all of our tasks, we acknowledge that this is a practical result. To practically ensure this, we have explicitly designed the coarse codes to have a limited number of codes (N=8; modeling the trend pattern) compared to the more large use of code size (N=256, 512), making sure that there is a higher chance of using all codes in the coarse codebook to map the trend pattern. However, we understand this is a practical outcome and would like to include it in the limitation section.
>
> Q6. EMD was necessary, as different codes may contain similar semantics. For instance, if code 1 and code 2 capture similar information, the transition between 1->1 and 1->2 should be deemed similar. Other metrics such as MSE cannot take this semantics into account (they treat 1 and 2 differently), while EMD can incorporate such information with the use of cost matrix M (defined as similarity in Eq 6). We direct the reviewer to the section "Channel Alignment via Optimal Transport, where we have provided a detailed example and the motivation behind the use of EMD.
>
> **Q7. Eval. protocol.** We direct the reviewer to our response **E1 of Reviewer n6fN (3rd reviewer)**

---

### Official Review · Reviewer_n6fN · 2025-03-14

**Overall Recommendation:** 3

**Summary:**

The paper introduces **TransPL**, a novel unsupervised domain adaptation (UDA) strategy for time series classification. It addresses the key challenge of domain variability in time series data arising from **temporal transitions** and **sensor characteristics**. To tackle this, the method employs a **coarse-to-fine VQ structure** to construct class- and channel-wise transition matrices. Additionally, the integration of **transition matrices** and **channel alignment** enhances both adaptation performance and interpretability. Notably, TransPL can seamlessly extend to weakly supervised scenarios and outperform existing pseudo-labeling methods in UDA tasks.

##  update after rebuttal

The authors explained the performance gaps between the results presented in the manuscript and those of prior work. This is due to their choice of interpretability over performance, which seems reasonable but also raises concern about whether such performance sacrifice is worthwhile. Overall, I think introducing the idea of vector quantization to time series adaptation is interesting and adjusted my score slightly.

**Claims And Evidence:**

The author emphasizes the significance of characterizing temporal transitions in time series data and integrating selective sensor (channel) shifts in time series domain adaptation task, and illustrates this concept with an example from human activity recognition.

**Essential References Not Discussed:**

The most recently developed methods on the topic are missing, e.g.,

[1] Dwlr: Domain adaptation under label shift for wearable sensor. IJCAI24.

[2] Caudits: Causal disentangled domain adaptation of multivariate time series. ICML24.

[3] Boosting Transferability and Discriminability for Time Series Domain Adaptation. NeurIPS24

**Experimental Designs Or Analyses:**

- Unjustified Performance Gaps with Baseline Methods:
  The paper adopts the same domain pairs and source risk selection strategy as ADATime (a 1D-CNN-based method), yet reports substantially worse performance (e.g., CODATS accuracy of 39.6 vs. 72.67 on the HHAR dataset’s 0-to-6 task), despite using a stronger backbone (patch transformer). A performance drop of ~30 percentage points across multiple adaptation tasks is alarming, especially since ADATime and Raincoat (reproducible via open-source code) have demonstrated reliable results.

  In short, why does the proposed method underperform ADATime and other reproducible baselines by such a large margin?

- Contradiction Between Model Complexity and Effectiveness: The patch transformer architecture is more complex than the 1D-CNN used in ADATime, yet it achieves significantly lower accuracy. This contradicts the expectation that advanced architectures should enhance performance.


    In short, can the proposed method improve performance when applied to simpler 1D-CNN backbones?

- Outdated and Incomplete Baseline Comparisons:

    The experiments exclude 2024 state-of-the-art methods, which weakens the paper’s effectiveness.

**Methods And Evaluation Criteria:**

* TransPL models the joint distribution $P(X,y)$ of the labeled source domain by constructing code transition matrices, which are then employed to pseudo-label the unlabeled target training set using Bayes' rule. However, the validity of interpreting temporal dynamic transformations using Markov chains remains open to debate.

* The paper evaluates TransPL on four widely used time series benchmarks, covering both human activity recognition (HAR) and electrocardiogram (ECG) classification.

**Other Comments Or Suggestions:**

The authors adopt an innovative approach to generate pseudo-labels for time series data, which enhances their accuracy and supports more effective training of target domain data. However, when deriving pseudo-labels for a specific target domain sample, leveraging both the source domain’s transition matrix and the target sample’s transition matrix in the computation could potentially yield slightly more precise pseudo-labels.

**Other Strengths And Weaknesses:**

* The proposed algorithm seems to exhibit considerable time complexity. Could the authors provide additional details?
* In the experimental section, the subsection titled "Use of Transition Matrix" lacks clarity and is somewhat difficult to follow. A more detailed or structured explanation would improve its comprehensibility.

**Questions For Authors:**

* Why does the proposed method underperform ADATime and other reproducible baselines by such a large margin?

* Can the proposed method improve performance when applied to simpler 1D-CNN backbones?

**Relation To Broader Scientific Literature:**

TransPL extends previous work by integrating ideas from UDA, pseudo-labeling, vector quantization, and optimal transport. A key highlight of the paper is the construction and application of the transition matrices.

**Theoretical Claims:**

The mathematical formulas in the paper seem correct, and the generation of pseudo-labels is based on methods from Bayesian inference and optimal transport.

---

> ### Author Rebuttal · Authors · 2025-03-29
>
> We first and foremost thank the reviewer for the thorough review of our work. We greatly appreciate your services. Here, we have prepared a detailed explanation to address each of the reviewer's questions.
>
> **E1. Performance of Baseline Methods.** Thank you for pointing this out. The performance gap stems from our use of patch transformer versus the 1D-CNN backbone in previous works. For fair comparison, we standardized all baselines with the same transformer architecture. While transformers typically require larger datasets to outperform simpler models in time series applications [1], our choice was deliberate: patch transformers preserve the semantic meaning of individual time segments (patch) in latent space, making coarse code transition matrices interpretable. **As we are trying to model the temporal transition in time series, having meaningful and interpretable patch representation is necessary for our whole methodology**. Unfortunately, 1D-CNN representations lack this interpretability as it mixes all features in the latent space. We acknowledge this design choice prioritizes interpretability of the constructed pseudo labels at some cost to overall performance, and we would like to include this in our limitation. Thank you for addressing this point.
>
> [1] Are Transformers Effective for Time Series Forecasting? [AAAI 22]
>
> **E2. Applying to 1d-CNN backbones**. Thank you for the suggestion. We believe that using 1d-CNN models could lead to enhanced model performance. However, as mentioned in **E1**, our methodology relies on the use of patch representation to keep the semantics. One possible way is to jointly optimize the 1d-CNN model alongside the patch encoder, but this would increase the computation, and also, **we would not be able to solely focus on the performance gain that can be brought with our coarse code transition matrix modeling.** As the whole paper focuses on developing pseudo labeling strategies that reflect temporal transitions and the channel-wise shift, the use of patch transformer was necessary.
>
> **E3. Need for additional baselines.** Thank you for this valuable comment. We acknowledge that some of the baselines were missing (CauDiTS did not release their code), however, **we have compared our work to 12 other more relevant baselines**. These baselines include relevant works such as RainCOAT (UDA) and T2PL (pseudo-labeling), which help us understand the significance of TransPL. While adding additional baselines could be helpful, we courteously request the reviewer to have a look at the additional analysis we have performed, which is novel and both needed to support our claims in the paper. For instance, we showed class-conditional likelihoods to
>
> **C1. Time Complexity.** The dominant computational cost comes from counting code transitions across all source samples. With N_s source samples, N patches per sequence, and D channels, this requires O(N_s × D × N) operations. However, implementation is practical due to (1) efficient vectorization techniques (da_utilities/transition_matrix.py file) for counting transitions and (2) the extremely limited number of coarse codes (only 8). For the UCIHAR dataset (9 channels, 14 patches per sequence), the total transition matrix construction time takes only 9.954±0.478 seconds.
>
> **C2. Use of Transition Matrix**. The purpose of this section was to highlight the benefits of using transition matrices to construct pseudo labels instead of directly constructing classifiers (1d-CNN, GRU, LSTM) that could be used to predict the pseudo labels. These discriminative models were trained using the (source) coarse code sequences and were trained to predict source classes. However, we show that such discriminative approaches fall behind our use of the generative approach (modeling the conditional likelihoods using the transition matrix), validating the use of transition matrices. Based on the reviewer's suggestion, we will refine this section for better clarity in the updated manuscript.
>
> **Q1,2.** We believe that we have addressed the questions in E1 and E2.
>
> We hope we have addressed the reviewers' concerns. Please let us know if any points require further clarification.

---

> > ### Comment · Reviewer_n6fN · 2025-04-07
> >
> > Thanks for the clarification. But I am now concerned about the setting of replacing 1D-CNN backbone with patch Transformer for all baseline methods. Is this a fair setting? From my side, it is meaningful to adopt patch Transformer for the proposed method since it aims to preserve the semantic meaning, but the baseline methods are not claimed to be designed for this purpose in their original papers. Also, there is no clear evidence that patch Transformer is inferior to 1D-CNN on small datasets or at least it cannot account for such large performance gaps.

---

> > > ### Author Response · Authors · 2025-04-08
> > >
> > > We thank the reviewer for the thoughtful feedback and acknowledge the valid concern regarding our experimental setup. We appreciate the opportunity to clarify our methodological decisions.
> > >
> > > **Justification for Using Patch Transformer:**
> > >
> > > 1. **Algorithmic Contribution vs. SOTA Performance** Our work's primary contribution is algorithmic—we explicitly model temporal transitions and selective channel shifts, which cannot be adequately represented with a 1D-CNN architecture. 1D-CNN inherently mixes information between channels and temporal features, making it impossible to construct the code transition matrix that is central to our proposed method.
> > > 2. **Semantic Preservation Requirement** As the reviewer acknowledged, PatchTransformer maintains the semantics of each temporal patch, enabling meaningful transitions between these patches. In contrast, transitions between latent representations from 1D-CNN lack interpretable meaning for our specific approach.
> > > 3. **Consistent Comparison Framework** We implemented all methods (our proposed method and baselines) using the same backbone **to ensure a fair comparison focused on algorithmic contributions** rather than architectural advantages. This isolates the impact of our domain adaptation techniques from backbone-specific benefits. To ensure fair comparison, we have also tested on the same 10 source-target pairs as in AdaTime and reported all results, while several later works only partially use the source-target pairs suggested in AdaTime.
> > >
> > > **Regarding Baseline Implementations:**
> > >
> > > For baseline methods, we have adapted popular algorithms that are **model agnostic domain adaptation algorithm** and **pseudo label algorithms** (DeepCoral, MMDA, SoftMax, NCP, SP, ATT, SHOT, T2PL) to work with the Patch Transformer backbone while preserving their core algorithmic contributions. None of the algorithms above claim to work on specific model architectures. While Raincoat and CoDATS were built on top of 1D-CNN, they do not leverage any specific properties of the 1D-CNN architecture in their adaptation mechanisms. Their core contributions, the loss functions and adaptation techniques, are entirely independent of the backbone choice.
> > >
> > > This standardization approach aligns with established practices in the field. As noted by AdaTime [1]: *"standardizing the backbone network choice is necessary to compare different UDA methods fairly. However, some previous TS-UDA works adopted different backbone architectures when comparing against baseline methods, leading to inaccurate conclusions."* By maintaining architectural consistency across all evaluated methods, we allow for a direct assessment of each UDA and Pseudo Label algorithm to domain adaptation performance. We believe our approach of using the same Patch-based backbone represents the fairest possible comparison given the requirements of our approach.
> > >
> > > We respectfully request the reviewer for an alternative experimental setup that would provide an equitable comparison while respecting the semantic preservation (our algorithmic contribution relies on this) requirements of our method. We would be grateful for this guidance and are willing to conduct additional experiments. We once again thank the reviewer for their dedication to reviewing our work.
> > >
> > > [1] AdaTime: A Benchmarking Suite for Domain Adaptation on Time Series Data
> > >
> > > Best,
> > >
> > > Authors

---

### Official Review · Reviewer_H1Hj · 2025-03-17

**Overall Recommendation:** 2

**Summary:**

This paper introduces TransPL, a novel pseudo-labeling approach for unsupervised domain adaptation in time series data. The authors argue that traditional pseudo-labeling strategies fail to capture temporal patterns and channel-wise shifts between domains, leading to sub-optimal pseudo labels. To address this, TransPL leverages vector quantization to model the joint distribution of the source domain through code transition matrices. The method constructs class- and channel-wise transition matrices from the source domain and uses Bayes' rule to generate pseudo-labels for the target domain. The authors claim that TransPL outperforms state-of-the-art methods in terms of accuracy and F1 score, while also providing interpretable insights into the domain adaptation process.

**Claims And Evidence:**

The paper claims that traditional pseudo-labeling strategies fail to capture temporal patterns and channel-wise shifts, but it does not provide sufficient evidence or analysis to support this claim. The motivation for why existing methods fail is not well-justified, and the paper lacks a clear explanation of why the proposed method is better suited to address these issues.

**Essential References Not Discussed:**

N.A.

**Experimental Designs Or Analyses:**

Lack of Ablation Studies on Coarse and Fine Codes: The paper introduces the concept of coarse and fine codes but does not provide ablation studies to show the individual impact of each. Without experiments that isolate the effects of coarse and fine codes, it is difficult to assess their respective contributions to the model's performance.

**Methods And Evaluation Criteria:**

The main concern in this part is the lack of Justification for VQVAE. Specifically, the paper does not provide a clear justification for why VQVAE is necessary or superior to other methods. The use of VQVAE feels like a technical add-on rather than a well-motivated choice, making it difficult to understand the true contribution of this component to the overall performance.

**Other Comments Or Suggestions:**

N.A.

**Other Strengths And Weaknesses:**

Strong Points:
1.	**Strong Experimental Results**: The paper demonstrates significant improvements in accuracy and F1 score across multiple time series benchmarks, outperforming existing state-of-the-art methods.
2.	**Clear and Well-Presented Visualizations**: The paper includes well-designed figures and tables that effectively illustrate the method's performance and interpretability.

Weak Points:
1.	**Lack of Justification for VQVAE**: The paper does not provide a clear justification for why VQVAE is necessary or superior to other methods. The use of VQVAE feels like a technical add-on rather than a well-motivated choice, making it difficult to understand the true contribution of this component to the overall performance.
2.	**Overly Complex Technical Framework**: The paper combines multiple advanced techniques, including VQVAE, optimal transport, and pseudo-labeling, without clearly isolating the contribution of each component. This makes it challenging to determine which part of the framework is responsible for the performance gains, and whether the method is overly complex for the problem at hand.
3.	**Lack of Ablation Studies on Coarse and Fine Codes**: The paper introduces the concept of coarse and fine codes but does not provide ablation studies to show the individual impact of each. Without experiments that isolate the effects of coarse and fine codes, it is difficult to assess their respective contributions to the model's performance.
4.	**Unclear Motivation**: The paper claims that traditional pseudo-labeling strategies fail to capture temporal patterns and channel-wise shifts, but it does not provide sufficient evidence or analysis to support this claim. The motivation for why existing methods fail is not well-justified, and the paper lacks a clear explanation of why the proposed method is better suited to address these issues.

**Questions For Authors:**

N.A.

**Relation To Broader Scientific Literature:**

Overly Complex Technical Framework: The paper combines multiple advanced techniques, including VQVAE, optimal transport, and pseudo-labeling, without clearly isolating the contribution of each component. This makes it challenging to determine which part of the framework is responsible for the performance gains, and whether the method is overly complex for the problem at hand.

**Theoretical Claims:**

N.A.

---

> ### Author Rebuttal · Authors · 2025-03-29
>
> We sincerely appreciate the reviewer’s thorough evaluation of our manuscript and the valuable feedback provided. We are pleased that the reviewer found our work to have strong experimental results and well-presented visualizations. Below, we have prepared a detailed response to address the reviewer’s concerns.
>
> **W1. Justification for VQVAE.** We respectfully disagree that VQVAE is merely a technical add-on—it is the cornerstone of our approach, which enables the density estimation of the source (**which is infeasible without VQVAE, Sec 4. Source Training**)
>
> VQVAE enables learning meaningful discrete representations of time series segments, which is essential for modeling the proposed temporal transitions in TransPL. To provide a very simplistic example, in a sequence like 1,2,5,6,1,2, VQVAE **learns** to map similar patches to discrete codes (1,2→"A", 5,6→"B"), creating a simplified and abstracted transition sequence A→B→A. Consequently, the use of transition matrices relies on these code transitions, and our proposed class-conditional likelihood generation for pseudo-labeling and the quantification of channel-wise shift rely on these transition matrices. As such, VQVAE is essential to our work. Other methods like SAX rely on predefined quantization schemes that may not capture data-specific patterns as in VQVAE. We would like to address the reviewer to Section 2.3 for detailed reasons behind our use of VQ.
>
> **W2. Design Principles** We respectfully note an oversight in this assessment. While the design may appear complex initially, each component serves a critical function in our framework. These elements work together to achieve our stated goal: effective time series pseudo-labeling that accounts for both temporal transitions and channel-wise shifts—the core motivations we established at the introduction.
>
> **(VQVAE)** - Please kindly refer to **W1**.
>
> **Optimal Transport (OT)** - TransPL requires comparing coarse code transition patterns between source and target domains while preserving semantic relationships. Unlike L1/L2 metrics that calculate absolute usage differences, **OT acknowledges semantic similarity between patches.** For example, if patches A and B encode similar information, transitions A→A and A→B should be considered similar. OT incorporates these semantics through a cost matrix defined by patch similarities, enabling a more meaningful comparison of transition patterns that respects the underlying data structure.
>
> Both modules are necessary designs for constructing pseudo labels, and we believe that they are not overly complex. They are the most simplistic method for obtaining pseudo-labels that account for temporal transitions and channel-wise shifts. No other pseudo labels take these into account.
>
> **W3. Lack of Ablation for Coarse Code.** We direct the reviewer to **Table 1**, which contains our detailed ablation study on the use of coarse codes. The first three rows evaluate single codebook performance with varying sizes, while the last four rows compare different codebook size combinations for coarse and fine codes. These results demonstrate that our proposed design principle yields the best pseudo-labeling accuracy.
>
> **W4. Unclear Motivation.** We respectfully disagree that our motivation is unclear. As we have noted in our Introduction, **our motivation is to design a pseudo-label method that considers the temporality and channel-wise selective shifts that happen in time series adaptation**. No other pseudo label methods consider both problems nor incorporate these into the pseudo labeling process of time series. Other pseudo-labeling methods (SoftMax, NCP, T2PL) rely on the clustering ability, representation performance of the source classifier, and without any focus on time series characteristics, while our methodology focuses on the two most important aspects of time series (time and channel). We have shown in Table 2 that other method fails to obtain strong adaptation performance, and that pseudo label accuracy is low compared to ours in Table 3. We also show that channel-wise alignment distance is better captured through our proposed work. We believe that we have sufficiently addressed our motivation throughout the whole paper, and the experiments conducted are well justified to back our motivation.
>
> We hope that we have addressed the concerns of the reviewers. If there is anything that is still unclear, please kindly let us know.

---

### Official Review · Reviewer_fkT4 · 2025-03-23

**Overall Recommendation:** 4

**Summary:**

The authors propose a new method, TransPL,  for unsupervised time series adaptation. Unsupervised domain adaptation deals with settings where a model trained on source domain  data with available labels has to be adapted to perform well in target domain with shifted  data  without available labels.

The proposed method uses vector quantized variational auto encoder (VQAE) to get discrete latent code books for time series patches. This VAQE model is pretrained on source data. A classifier is also trained on top of these source encodings using the available source labels.  The learned discrete code books are used to obtain transition matrices which describe how a time series patterns switch.
The authors use this scheme to obtain transition matrices for each channel.
 The  computed transition matrices for the source and target domain channels are used in optimal transport distance to obtain domain discrepancy scores for each channel. This scheme ensures that the  computed optimal transport distance, and thus the resulting channel score, is lower for channels with large shifts between source and target domains.
These  channel scores are used as   weights for channel specific  posteriors for each  class. This  weighted (and normalized) sum across all channels of these channel specific posterior provides the posterior probability for the a particular class. This is repeated for all classes to obtain posterior score across all classes for the provided input.
These  posterior scores for a target data point can be used to obtain pseudo labels for unlabelled target domain data.

High confidence pseudo labels in the target domain are used to to optimize the source model( VQAE encoder, decoder, codebooks and classifier on top of the encodings)

The proposed method is able to selectively down-weigh channels that have  a large discrepancy. It can also incorporate different label probabilities as priors when computing posterior probabilities. This allows the method to incorporate weak target supervision when target label distribution is available.

Extended results on multiple domain adaptation datasets show that the proposed method improves performance on existing domain adaptation methods.

**Claims And Evidence:**

The claims made in the paper are mostly supported by convincing evidence.

The authors show that the proposed method results in improved performance over existing methods.

The ablation studies  show how their proposed method's ability to incorporate weak supervision and  adaptive channel alignment improve performance. These ablations provide clear and convincing evidence to support the stated claims.


The authors also provide ablation studies for the proposed channel adaptation layer, but visualizations on a single 3 dimension example are not fully convincing to back their claim the proposed scheme of capturing scheme and discarding channels with large shifts. This significance of these claims is also limited  by the  ablation results  in Table 5, where we see that the PTB dataset, with 15 channels, has its performance degraded when channel adaptation is used. Maybe this is a misunderstanding on my end, but for a dataset with large number of channels, the channel adaptation strategy should further improve performance.

I would suggest a more thorough experimental analysis to verify that the proposed channel adaptive layer and discard channels with large shifts. This can be done through simulated examples on existing datasets where random channels are corrupted through noise.


The authors also claim that the proposed methods provides explainable insights for domain adaptation. This is slightly vague, and I can't seem to find direct experiments which back this claim.

**Essential References Not Discussed:**

The authors mostly discuss all essential references.
They do claim that existing domain adaptation methods can not model channel level shifts.

There has been very recent work [1] that tackles channel level shift for unsupervised domain adaptation.
This paper seems to have been published pretty recently and perhaps before the ICML deadline, so ofcourse its totally understandable for the authors to not mention this originally.

Though given how relevant this work is, mention [1] would strengthen the proposed work section.



[1] Ahad, N., Davenport, M. A., & Dyer, E. L. Time Series Domain Adaptation via Channel-Selective Representation Alignment. Transactions on Machine Learning Research.

**Experimental Designs Or Analyses:**

I have checked the experimental design and analysis results provided in Table 1 (that provides VQAE code performance interns of data reconstruction, and pseudo labelling accuracy).

I also checked experimental design results for Table2 (adaptation performance on target), Table 3 (accuracy of pseudo labels), and Table 4 (weak supervision UDA), and ablation studies Table 5.


These experimental design make sense.

**Methods And Evaluation Criteria:**

The proposed method and evaluation criteria mostly  make sense.
The authors test on commonly used benchmarks for time series adaptation. They also provide a thorough comparison with existing unsupervised domain adaptation methods.
They also provide use commonly used metrics such as target domain AUC/F1 score to evaluate unsupervised domain adaptation performance

They also show how their proposed method improves pseudo labelling accuracy as compared to other baseline methods that also employ pseudo labelling.

Additional results are also provided which show how the performance degrades when their proposed approach of modeling both coarse and fine codes through VQAE is replaced by directly modeling codes through other encoders such as 1DCNN.

**Other Comments Or Suggestions:**

The paper is mostly well written and I wasn't able to find any  glaring typos

**Other Strengths And Weaknesses:**

Strengths:

- A promising  scheme of incorporating transition matrices for domain adaptation. This is very novel contribution which hasn't been explored within the context of unsupervised time series adaptation
-  A scheme which allows to incorporate weak supervision
- Thorough evaluation on datasets and comparison with numerous  to support claims made.


Weakness:

- There is not discussion on limitations of their proposed work. Unsupervised domain adaptation practically only works if many assumptions are met. [2] provides an overview of these assumptions for images, but the same assumptions can be extended to shifts for any types of domains.
 It makes sense to support their claim that code transition matrices are more invariant across changes (and can also hep identify and ignore channels with large shifts), but there certainly can be cases where the domain shift is large that transition matrices from incorrect classes are nearby.  There is no one domain adaptation method which is best for all scenarios. In some cases Frequency information might be more invariant,(like proposed by RAINCOAT), but in other scenarios transition matrices might be more invariant .
- The analysis on channel alignment and how it affects performance in section 7.6 is weak and only considers one example of a 3 channel dataset. Considering how performance with channel alignment does not improve significantly on the  15 channel PTB dataset, I would suggest more through experiments to study how the proposed method effectively ignores channels with large shifts. This can be done through inducing channel level corruptions, and analyzing the proposed model's performance in terms of accuracy of target domain data , as well as the ability to ignore corrupted channels.



[2] Gijs van Tulder and Marleen de Bruijne. Unpaired, unsupervised domaadaptation assumes your domains are
already similar. Medical Image Analysis.,  2023.

**Questions For Authors:**

- How is the cross entropy layer classifier head trained? is traded on top of coarse codes? This was perhaps not clear from the text, or Figure 1 in its present form
- How are class wise class conditional likelihoods obtained using class wise Transition matrices? This is perhaps not clear in the text in its present form
- - There can be scenarios where code transitions should be computed across multiple channels as compared to once channel. Would these channel specific coarse codes capture correlations across channels which could be important to capture?
- Can there be cases where there is a large shift between channels, but the  transition matrices are still relatively close for channels? E.g there might be a large DC shift in the channel,
- The mutual information between channel content and class labels can vary across channels. There certainly can be cases where class information is mostly contained in only a few channels. how would the proposed method perform when there is a large corruption in such channels across source and target domains. Would the proposed scheme ignore such channels (as the transition metrics could be very different leading to lower $w_d$?) This is ofcourse totally fine, but this could be a limitation that needs to be made explicit for practitioners and readers.

**Relation To Broader Scientific Literature:**

The authors sufficiently place their paper in context of the broader scientific literature for domain adaptation and learning discrete latent codes.

**Theoretical Claims:**

There are no theoretical claims made in this paper

---

> ### Author Rebuttal · Authors · 2025-03-29
>
> We sincerely thank Reviewer fkT4 for the thorough, insightful, and constructive feedback. We appreciate that the reviewer has found our work a novel contribution for UDA time series that has not been explored elsewhere. We also thank the reviewer for thinking through our problem definition with us. We have provided detailed responses to address each of its comments.
>
> **Q1. Training Cross-Entropy.** The cross-entropy is calculated using a [CLS] token appended to the input patches. After processing through the Encoder, this [CLS] token—having attended to all time series tokens via transformer—is used to train the classifier. We will add a guiding arrow in **Fig. 1** to clarify this flow. We also refer to Lines 194-196.
>
> **Q2. Class Conditional Likelihood.** After encoder training, we process unlabeled target time series to obtain discrete code sequences (e.g., A-B-A). We then calculate the likelihood of observing this sequence under each class-specific transition matrix from the source domain, similar to maximum likelihood estimation in Hidden Markov Models. For instance, for a given sequence A-B-A, the Class 1 Transition matrix may output higher likelihood, and the Class 2 Transition matrix may output low likelihood. This allows us to assign the most probable class label to each target sequence. We direct the reviewer to **Eq. 7** and **Fig.1** for additional clarification.
>
> **Q3. Correlations between Channels.**
>
> Currently, we use the same codebook to model all sequences in a time series but construct channel-wise transition matrices to represent each channel. As such, we could presumably say that the shared codebook can capture correlations between channels. While we have also tried modeling each channel with separate codebooks, we empirically found that this did not lead to better pseudo-labeling results (It might be due to increased complexity; alternative design methods should be further looked into).
>
> **Q4. Transition Matrix Remains Similar.** This is also an interesting point. We believe that in such a scenario (e.g., DC shift), the transition patterns would greatly differ, as our codebook does not solely focus on the shapes of the time series pattern. If a large DC shift occurs, it would most likely return different time series signal values that should be reflected in the coarse code usage pattern.
>
> **Q5. Channel Importance.** We find this really interesting and appreciate this insightful point. The reviewer is correct that only certain channels may contain class-relevant information [1], and when class information is concentrated in channels experiencing significant domain shift, our method would assign lower weights to these channels due to the large EMD distance. This is a limitation we'll explicitly acknowledge in our revision. We plan to extend our approach in future work by incorporating channel importance measures alongside distribution shift metrics, allowing for weighting that considers both channel-shift and the importance of channel for classification.
>
> [1] CAFO: Feature-Centric Explanation on Time Series Classification (KDD'24)
>
> **R1. [TMLR paper]** We will include the provided reference. We also think it will strengthen our work!
>
> **W1. Limitation**. As the reviewer has noted in Q5, we believe that TransPL may not operate optimally in cases where the heavily shifted channel contains class-discriminative information, as our pseudo-labeling method would weigh less on such channels. We will incorporate this into our Limitation Section in the updated manuscript.
>
>  **W2. Additional Analysis.** We appreciate the reviewer's valuable insight regarding channel alignment analysis. We propose the following experiment to address this concern:
>
> **Analyzing Channel-Level Corruption.** We plan to inject random noise into high-weight channels ($w_d $) in the adaptation scenario and measure adaptation performance by tracking the reduction in $w_d $ after noise injection. A reduction would demonstrate the model's ability to downweight corrupted channels. Would the reviewer consider this a suitable approach to provide more thorough evidence of our method's capability to identify and ignore channels with large distributional shifts?
>
> We once again thank the reviewer for providing helpful feedback to improve our work. Please let us know if there are any additional uncertainties so that we can address them. Thank you.

---

> > ### Comment · Reviewer_fkT4 · 2025-04-05
> >
> > Thank you for your detailed answers to my questions.
> >
> > Also, thank you for the clarification to my questions on cross entropy and class conditional likelihood.
> >
> >
> > These additional  experiments such as analyzing Channel level corruptions, channel importance would certainly strengthen the paper.
> >
> > I also think it might be worth including experiments which do test whether the transition matrix remains similar across diffrent types of shifts. This would help readers better understand the limitations or scenarios where transition matrix could vary significantly across domains.
> >
> >
> > One additional question: Do you think is approach is scalable as the number of dimensions increase? As that would involve computing different transition matrices for each channel? Would this be scalable when there are  100 channels?

---

> > > ### Author Response · Authors · 2025-04-06
> > >
> > > Dear Reviewer **fkT4**,
> > >
> > > We sincerely thank the reviewer for the thoughtful feedback on our work. We're pleased that our clarifications regarding cross entropy and class conditional likelihood have addressed the reviewer’s earlier concerns. Below, we provide detailed responses to the remaining suggestions.
> > >
> > > **Channel-Level Corruptions** Following the reviewer’s recommendation, we conducted channel-level corruption experiments on the UCIHAR task across all 10 source-target pairs. (We provide the experimental results through an Annonymous Github). After analyzing the data, we identified that the 6th channel consistently exhibited high $w_d$ values across most of these pairs. When we introduced increasing levels of noise (Gaussian) to this specific channel, we observed a corresponding decrease in $w_6$ values, confirming the utility of Channel-Level Adaptation. We will incorporate these experimental results and their analysis in our revised manuscript. The reviewer’s suggestion has significantly strengthened our empirical validation.
> > >
> > > Experimental Results: https://anonymous.4open.science/r/TransPL_Anon2-C325/
> > >
> > > **Different Types of Shifts** We appreciate the reviewer’s recommendation to examine transition matrices across various shift types. This is indeed an intriguing direction. However, we face challenges in determining appropriate shift types and methodologies for synthetically inducing diverse shifts in our time series data. We would greatly appreciate it if the reviewer could suggest relevant literature that might guide our approach to this question. We're eager to explore this direction in our ongoing work.
> > >
> > > **Scalability to Increased Channel Dimension** Our approach demonstrates strong scalability with increased channel dimensions, as the transition matrices between limited coarse codes remain computationally efficient. While higher channel dimensionality poses less significant challenges to our TransPL framework, we acknowledge a practical tradeoff between the number of patches and the computational time required to count transitions between these patches. For time series problems with more than 100 channels, we would suggest first filtering and reducing the number of channels to those most relevant to the task at hand. TransPL can then be effectively applied to these filtered time series, maintaining computational efficiency while preserving performance.
> > >
> > > We once again thank the reviewer for their constructive feedback, which has significantly strengthened our work. We remain available to address any additional questions or concerns.
> > >
> > > Best Regards,
> > >
> > > Authors.

---

### Decision · Program_Chairs · 2025-05-01

**Decision:**

Accept (poster)

**Comment:**

This paper presents TransPL, a novel pseudo-labeling approach for unsupervised domain adaptation in time series data. To effectively capture temporal dynamics and channel-wise distribution shifts across domains, TransPL employs vector quantization to model the source domain's joint distribution via code transition matrices. Specifically, TransPL constructs class- and channel-wise transition matrices from the source data and applies Bayesian inference to generate high-quality pseudo-labels for the target domain. Experiments show that TransPL surpasses state-of-the-art methods in both accuracy and F1-score while offering interpretable insights into the domain adaptation process.

During the discussion, the rebuttal well addressed prior concerns regarding the justification for using VQVAE and optimal transport. The AC agrees with the two positive reviewers that while state-of-the-art (SOTA) performance and marginal improvements are worth noting, they should not be the primary focus—given that the authors introduce a novel and conceptually compelling idea to the field.